# A Geotechnical Perspective on a Complex Geological Environment in a High-Speed Railway Tunnel Excavation (A Case Study from Türkiye)

Candan Gokceoglu [1],*, Ebu Bekir Aygar [2], Hakan A. Nefeslioglu [3], Servet Karahan [4] and Suat Gullu [4]

1   Department of Geological Engineering, Hacettepe University, Beytepe, Ankara 06800, Türkiye
2   Fugro Sial Geosciences Consulting Engineering Co., Cankaya, Ankara 06680, Türkiye
3   Institute of Earth and Space Sciences, Eskisehir Technical University, Tepebasi, Eskisehir 26555, Türkiye
4   TCDD, General Directorate of Turkish Railway System, Altindag, Ankara 06050, Türkiye
*   Correspondence: cgokce@hacettepe.edu.tr or candan.gokceoglu@gmail.com; Tel.: +90-5324730851

**Abstract:** The T26 tunnel was designed within the scope of the Ankara-Istanbul high-speed railway in accordance with the speed of 250 km/h. Some serious problems and excessive deformations were encountered during the excavation works. The deformations in the tunnel caused subsidence on the surface and the Tunnel Boring Machine (TBM) became stuck; therefore, tunnel excavation works were suspended. Design works for re-excavation in the T26 tunnel and extracting of the TBM were carried out and the tunnel was re-designed by the New Austrian Tunneling Method (NATM) system. The main purposes of the present study are to describe the problems encountered during the T26 tunnel and to discuss the sources of the problems. The advantages and disadvantages of TBM and NATM methods for the tunnel having difficult ground conditions were discussed. Critical points needing to be considered for the tunnels excavated with TBM through weak ground conditions and the effect of the TBM selection process were discussed. Considering the complex geological and geotechnical structure of the tunnel route, it is possible to say that the T26 case is an interesting case for tunnel engineering. Along the tunnel route, landslides, high seismic activity, groundwater conditions, and extremely weak rock mass features coexist. Therefore, the tunnel route is a very complex environment. However, due to the geometric limitations of the high-speed railways, relocation of the route is not possible. The experiences gained from tunnel excavations under difficult conditions are capable of bringing new horizons to future tunneling studies.

**Keywords:** high-speed railway tunnel; landslide; TBM; NATM; numerical analyses; weak rock mass

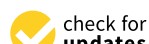



## 1. Introduction

In today's tunneling studies, long tunnel excavations with Tunnel Boring Machines (TBM) are preferred because of speed and cost. However, there are two important factors for the success of TBM tunneling. The first one is to select the appropriate TBM and the other one is to coordinate properly the TBM excavation process during site works. To select a suitable TBM, the geological and geotechnical conditions of the study area must be determined accurately, at first. For this purpose, geological units and structures should be identified utilizing surface engineering geological surveys and detailed borehole investigations. Unfortunately, depending on those surveys, especially for long tunnels planned to be excavated by TBM, it is hard to define the geological and geotechnical situations of the study area accurately. According to Barton [1], major, and sometimes seemingly minor fault zones represent the 'Achilles heel' of TBM. Therefore, some serious problems are faced due to the lack of accurate geological and geotechnical information on the tunnel route. However, for rock masses with extremely low Q-values, TBM performance can be estimated using the approach proposed by Barton [2]. When selecting the type of TBM for long tunnels to be excavated through hard rock conditions and situated beneath

a large overburden thickness, open gripper TBM may result in serious problems if fault zones, and weathered units are encountered [1]. Additionally, in case of unexpected ground conditions, it is not possible to make instant modifications after starting excavations. Therefore, even though tunnels excavated by TBM seem to be fast and economical, the method bears severe risks. Hence, choosing a suitable TBM is mainly the most crucial factor for the success of an implementation. The other important factor is the operations such as probe drillings performed during excavation works to estimate geological conditions. Probe drillings are very important during excavation works. If a suitable TBM cannot be selected, problems encountered during tunnel excavation may cause the destruction of the TBM or serious time losses. Landrin et al. [3] revealed some of the impacts during tunnel excavation. It is seen that both time loss and material damage may arise from possible collapses during TBM excavations [3]. In literature, several studies on TBM jamming [4–10] were published. On the other hand, some detailed studies on the interaction between shield, ground, and tunnel support in TBM tunneling through squeezing ground [11], and the influence of discontinuities on the squeezing intensity in high in-situ stresses [12] were investigated. As can be seen from these studies, one of the major problems for TBM is squeezing ground conditions. In 2013, the excavation was interrupted in the T26 tunnel after the occurrence of collapse and the TBM being stuck. Subsequently, no site work took place inside the tunnel for about 4 years. At the end of 2017, excavations were initiated once again with the New Austrian Tunneling Method (NATM). In the literature on tunnels, it is possible to find serious cases of problems with TBM implementations. The studies conducted for TBM tunnels excavated under difficult ground conditions were summarized by Zolfaghari and Mokhtari [13]. Within the scope of the India Kishanganga hydroelectric implementation, a double-shield TBM was used during the excavations of the tunnels constructed at the foothills of Himalayan mountains under difficult ground conditions with a length of 14.6 km and a diameter of 6.18 m [14]. The tunnel excavation was carried out in andesitic basalts of the Panjal formation and meta-siltstones of the Radjhan formation. During the excavation in the basalt and andesite units, collapses occurred in low cohesive units with clay infillings that have unsuitable discontinuity orientations in three main regions. To pass through these collapses, gaps were filled with grouting and locally with foam, and a bypass tunnel aiming to reach the TBM head was excavated. These studies resulted in a loss of approximately 5 months [14]. Miyazawa et al. [15] examined the problems experienced in the Kuriko tunnel situated between Fukushima and Yonezawa with a length of 9000 m and a diameter of 4.5 m. The tunnel was mainly excavated through hard granite, rhyolite, dacite, conglomerate, sandstone, and mudstone; for this reason, open-TBM was preferred. During the tunnel excavation, there were collapses in the ceiling section of the tunnel. To overcome this problem grouting was applied utilizing the installation of long umbrellas and in turn, the stability of the ceiling was to be secured [15]. The impacts of the 4.5 m diameter water transportation tunnel in Iran Ghomroud were evaluated by Zolfaghari and Mokhtari [13]. The tunnel, which has a total length of 18 km, was excavated with a double-shield TBM. It was excavated within the Jurassic-Cretaceous metamorphic and sedimentary lithologies between the Arabian and Iranian plates. During tunnel excavations, poor geological conditions, raised from the fault zones and weathered zones were encountered many times. In these parts, there were collapses, and the TBM was stuck [13]. In the study conducted by Lin and Yu [16], the problems in the Westbound (WB) Hsuehsan Main Tunnel which was excavated with double-shield-hard rock TBM were investigated. During the tunnel excavation having an 11.7 m diameter and 12.9 km length, while the Shanshin Fault was being passed, serious problems were encountered inside the tunnel. A period of 2 months was wasted to pass through the first collapse. After this process, the excavation works were continued with the TBM again. Subsequently, with only an advance of 26.5 m, the segments failed and a collapse on the TBM was experienced. In the meantime, serious water discharge occurred to the tunnel and the TBM was completely submerged in the water. Consequently, the water discharge ended up with a severe debris income causing the TBM to sink completely into the material. After this point, tunnel

excavation was continued by the drill and blast method [16]. Re-excavation of the section to remove the stuck TBM inside the Hsuehshan tunnel lasted 34 months. From this point of view, the aforementioned cases manifest similarities with the T26 tunnel in terms of the applied method, encountered problems, and suggested solutions. As valuable experience in engineering practice, the present study aims to describe the ground conditions of the T26 tunnel route and to evaluate the reasons for TBM excavation failure. Every tunnel built in complex geological conditions presents its own unique problems, and from this perspective, every tunnel case opens new horizons for tunnel engineering. Therefore, in the present study, the problems encountered during the T26 tunnel excavations are defined and possible solutions are discussed. The T26 tunnel route has extremely weak rock mass features, landslides, high seismic activity, and groundwater. Due to these features, various geotechnical problems were encountered during the T26 tunnel excavations and engineering solutions were suggested. In conclusion, this study is an important case in tunnel engineering with the complex geological features of the tunnel route.

## 2. T26 Tunnel

In the last decade, Türkiye has started to construct high-speed railways among the largest cities such as Ankara-Istanbul, Ankara-Sivas, and Ankara-Izmir. During these construction efforts, several tunnels have been constructed [17–20]. One of these tunnels is the T26 tunnel in the Ankara-Istanbul High-Speed Railway construction project. The T26 tunnel excavation was initiated in July 2009 with NATM, and then the construction method was changed to TBM, and the first 1020-m section was completed with TBM. Excessive deformations occurred during tunnel excavation and resulted in a settlement at the surface and failure inside the tunnel. Subsequently, many continuous reinforcement efforts were performed. Following the additional reinforcements, the TBM stuck after advancing 246 m more due to the occurrence of extreme deformations at Km: 217 + 526, and as a consequence, excavation works were interrupted. In 2016, the re-design stage was initiated by considering the NATM method instead of TBM. After completion of the re-design of the T26 Tunnel, the excavation was started in 2017 with the NATM. The tunnel route is located between Bilecik and Bozuyuk stations at Km: 216 + 260 and Km: 221 + 715, respectively. The entrance portal of the tunnel, which passes through steep topography, is located 1.7 km southeast of Baskoy town of Bilecik city while the exit portal is located 1.3 km northeast of Demirkoy town (Figures 1 and 2). The T26 tunnel, which was designed as a single-tube and double-lane, has a width of 13.30 m and a height of 8.0 m.

## 3. Geology of the Site Vicinity

The T26 route is located in northwest Anatolia, in the Izmir-Ankara Suture Zone, in the tectonic unit known as the Sakarya continent [21,22] or Sakarya zone [22,23]. The Izmir-Ankara suture represents part of the boundary between Laurasia and Gondwana along which a wide Tethyan ocean was subducted [24]. Due to the extremely complex geological structure of the region, the region was the subject of several geological studies [25–30]. The Sakarya zone is an area where the "Central Sakarya Base" is observed at the base [22,31,32] and various lithological units from the Lower Jurassic to Quaternary are observed at the top. The central Sakarya basement is composed of two major groups of rocks. The first group includes continental-origin metamorphic rocks including mica-schist and gneiss [33,34] and Permo-Carboniferous (290 Ma) aged [35] meta-granitic rocks. The other group of rocks is composed of a meta-basic, meta-pelitic sequence similar to mélange having a schistose structure in itself [22]. Meta-basics and meta-pelites and continental meta-granitic sequences are overlain by Lower Jurassic aged clastic rocks [22,31,32]. Post-tectonic sedimentation on the metamorphic basement starting from the Early Jurassic lasts until the Cretaceous-Early Eocene with some discontinuities, and the top cover units of the region consist of Eocene aged shallow marine deposits and Miocene aged lacustrine sediments [22,31,32]. There are complex-nature meta-pelite, meta-basic, and serpentinite units among the units; felsic

intrusive cut them with intrusive interaction; Miocene aged sediments, and alluvium is found in the region in which the T26 tunnel route is located (Figures 3 and 4) [22,36].

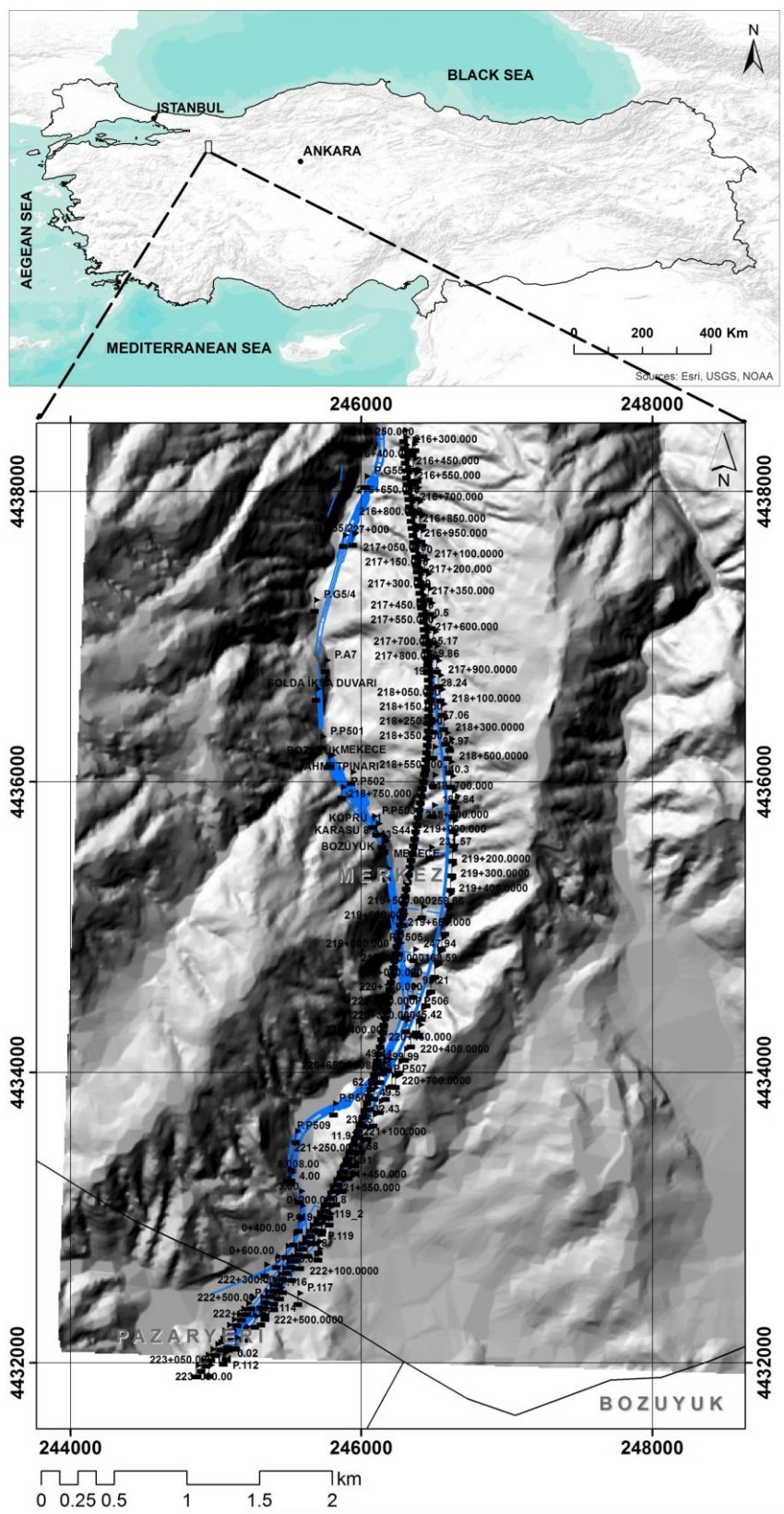

**Figure 1.** Location map of the T26 tunnel; the high-speed train route is indicated using the black line, and the highway is represented using the blue line.

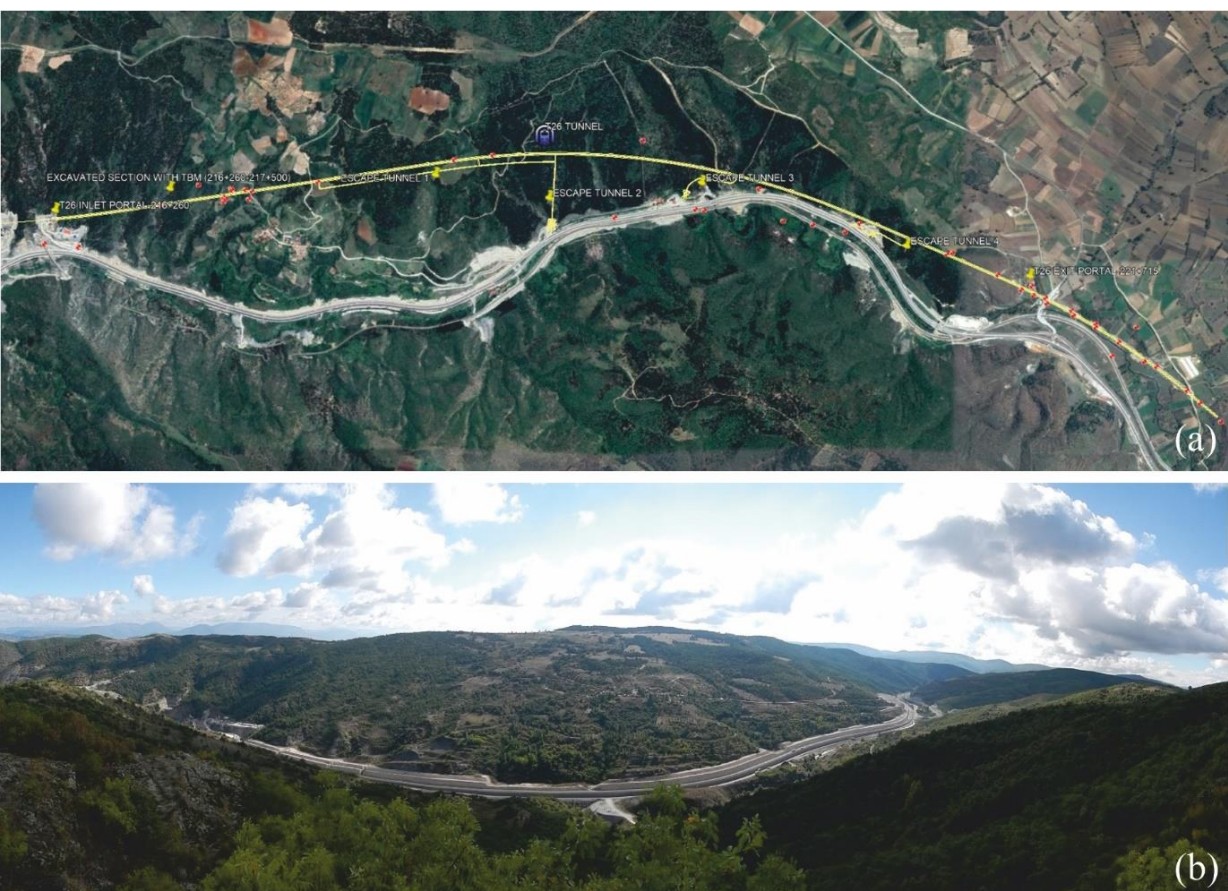

**Figure 2.** A view of the T26 tunnel route from Google Earth (**a**); a panoramic view from the site vicinity of the T26 tunnel (**b**).

The tunnel route is located in one of the highest seismic zones of Turkey. There are two main tectonic elements within the study area and its immediate surroundings. The first one of them is the North Anatolian Fault (NAF) located approximately 45–75 km north of the route and the other one is the Kütahya Fault passing through approximately 70 km south of the zone. Apart from these, within the route and its immediate surroundings, there is the İnönü Dodurga Fault Zone consisting of many small segments and located approximately 25 km south of the route and the Eskişehir Fault Zone located approximately 60 km southeast of the area. Consequently, these faults are active and produce significant earthquakes. One of the faults that can generate peak acceleration in the T-26 tunnel route is NAFZ.

The Armutlu Peninsula South Branch and an earthquake with a magnitude of 7.5 will be generated by it in a return period of approximately 120 years; this will generate a horizontal acceleration of around 200 gal along the route.

As can be seen from Figure 4, T26 excavations were started from the inlet portal and consisted of approximately 1000 m graphite schists from here. Later, after the 200 m metabasite zone, there are again approximately 500 m graphite schists. From here on, T26 was excavated mainly in chlorite schists, and in places fault zones, serpentine and weathering zones exist (Figure 4). Among these, the graphite schists are extremely poor rock masses showing heavy squeezing characteristics.

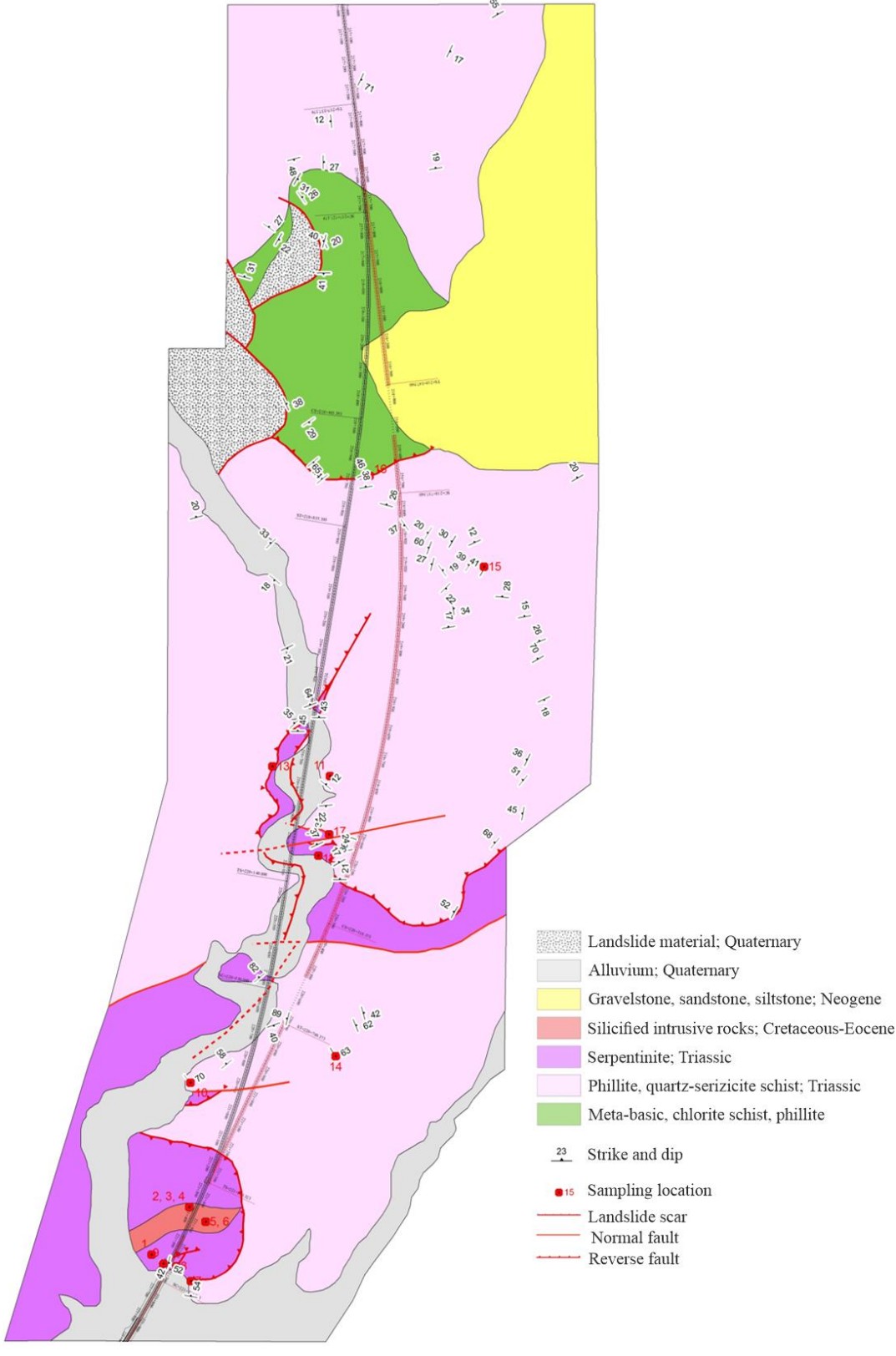

**Figure 3.** Geological map of the site vicinity; the primary design route is indicated using the purple line [22].

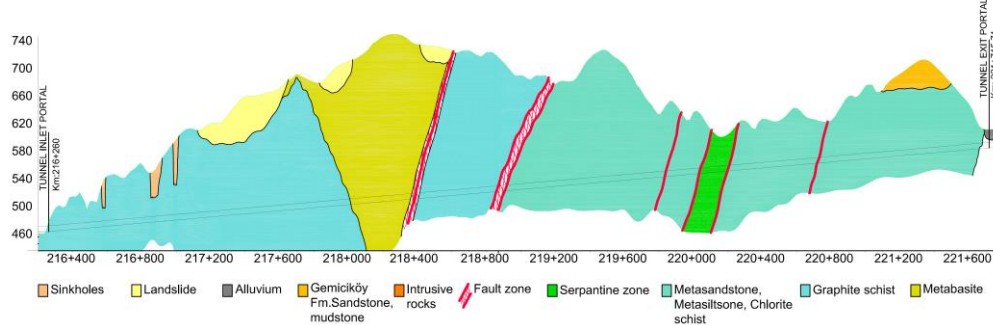

**Figure 4.** Longitudinal cross-section of the T26 tunnel [36].

## 4. Difficulties Encountered during Excavation

The first investigations were initiated in 2004 and the excavation works were started in 2009 and a portal excavation advancing through 30 m was completed. As the tunnel excavation passed through an active landslide, a landslide affecting a 100 m section of the tunnel route occurred and this section was blocked. Eventually, the tunnel excavation was stopped, and this section was abandoned (Figure 5). After the occurrence of this landslide, five boreholes with inclinometer installation were drilled near the portal. The inclinometer measurements revealed that there was a clear movement and the existence of a landslide. Therefore, the tunnel route was relocated 200 m away towards the mountainside to avoid the landslide. During these design works, in 2010, the excavation and tunnel support implementations were proposed depending on the NATM method, and site applications were initiated, accordingly. After an advance of 260 m inside the tunnel via the NATM method, to proceed faster, it was decided to progress with the application of the TBM method for the rest of the tunnel.

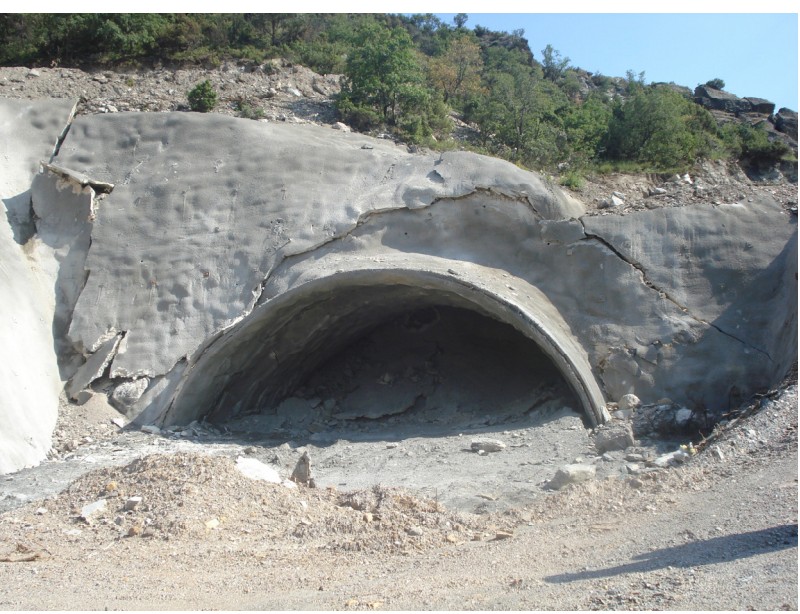

**Figure 5.** The entrance portal of the T26 tunnel abandoned in 2009 [36].

### 4.1. Excavation of the T26 Tunnel with TBM

In 2011, the excavation works were re-started by selecting a hard-rock TBM. This part of the tunnel ranges from Km: 216 + 260 to Km: 222 + 360 and has a total length of 6100 m. This section is a double-track TBM tunnel with 13.40 m external and 12.95 m internal diameters. The nominal excavation diameter is 13.77 m. The theoretical ring room (annulus) around the segments is 0.185 m. The theoretical excavation volume per meter advance is 149 m$^3$/m.

The theoretical grouting volume of the annulus is 15.8 m³ [37]. The lining consists of seven concrete segments and one keystone. The segment thickness is 0.45 m. Each segment has a 33 mm rubber gasket at the extrados side of the segments for waterproofing. In the area where the segment damage occurred, C40/50 concrete with 65 kg/m³ reinforcement (10/∅12) plus 25 kg/m³ fibers were used. After the design changed, to increase strength, the concrete of the segments was improved to C60/70. The reinforcement was increased to 125 kg/m³ and then to 225 kg/m³ as well. The steel diameter was also increased from ∅12 to ∅24 due to the deformations.

### 4.2. Settlements and Subsidence on the Surface along the Route

Throughout the TBM excavation, which used a hard rock TBM with an open mode between Km: 217 + 720 to 217 + 770 (see Figure 4), three subsidence developed with a depth of 25–30 m and a diameter of 35–40 m. (Figure 6a). These settlements were observed inside the tunnel during the excavations. In other words, a protective zone around the tunnel could not be established so a plastic zone reaching the surface and dead load was situated. Overburden thickness was reported as around 105 m at these sections [37]. After these problems, necessary modifications were ensured on the TBM; thereafter, the tunnel excavation proceeded. By converting the TBM type from hard rock to Earth Pressure Balanced Machine (EPBM), excavation was advanced for approximately 250 m. Afterward, no more subsidence was observed except for some fractures on the surface and slopes (Figure 6b).

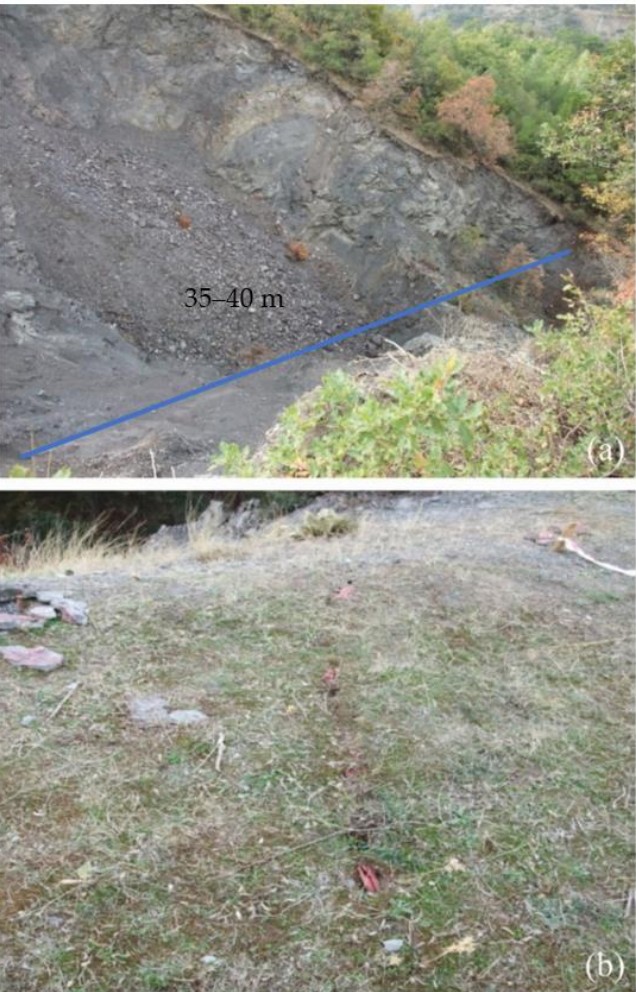

**Figure 6.** View of subsidence developed throughout the TBM excavations (**a**); tunneling induced surface cracks adjacent to the tunnel alignment (**b**).

*4.3. Failures and Fractures Observed on the TBM Segments*

During the excavation works, severe deformations (Figure 7) were observed at the segments. To prevent these deformations, "I" profile supports were installed at the damaged segments. These profiles were selected as NPI 200 steel rib type and installed with a round length of 50–100 cm. They were also reinforced with shotcrete application that was strengthened with fiber-concrete and wire mesh (Figure 7a,b).

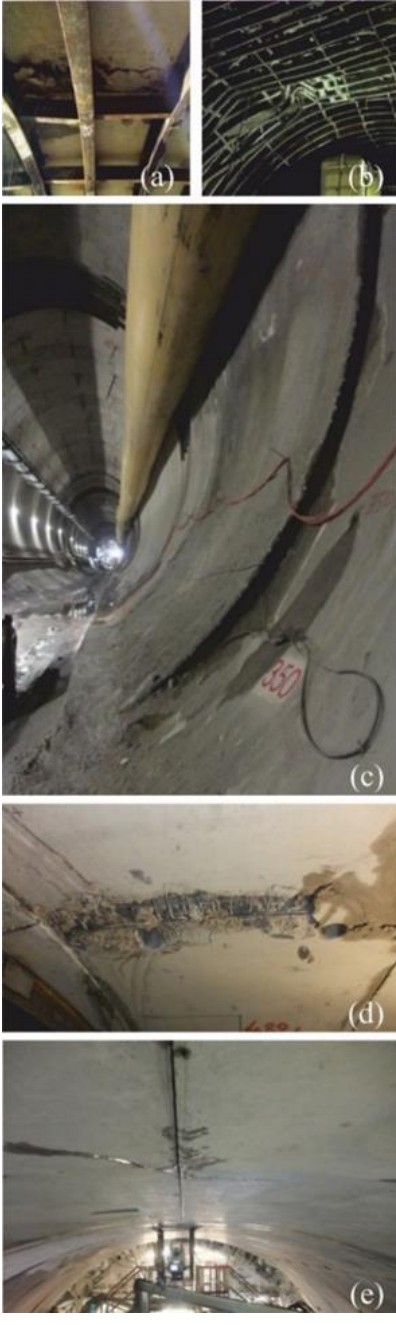

**Figure 7.** Damages on the segments and the reinforcement with NPI profiles (**a**,**b**); additional supports; steel-beam, wire mesh, and fiber-concrete shotcrete (**c**); an example photograph representing excessive plastic deformation in the tunnel (**d**); an example view of a longitudinal joint in the tunnel (**e**).

Daily progress was decelerated to 2.0 m after November 2012. To prevent damages that may cause a collapse in the excavated sections and ensure securing the tunnel stability, NPI profiles were installed with 1 m spacing. At this stage, fractures at the tunnel surface (Figure 7c–e), holes on road above the tunnel route, and slides were observed. Subsequently,

a report comprising opinions and recommendations was prepared by Mott MacDonald [38] to safely advance the excavations. Additionally, the Swiss Federal Institute [39] prepared a report analyzing squeezing problems in the tunnel.

### 4.4. Investigation of the Collapsed Zone and Squeezing Mechanism

The stability problems experienced by the tunnel face during the TBM excavation negatively affected the tunnel stability. The subsidence encountered during the excavation affected the process and caused yielding on the segments in the tunnel and caused the TBM to be revised. The cause of these problems was the failure of tunnel face stability. One of the most important factors affecting tunnel stability is tunnel face stability [40–46]. In addition, planning a TBM drive in squeezing ground is a complex problem for tunnel engineering [47]. For this purpose, 3D numerical analyses were carried out with the FLAC3D program [48]. Additionally, the stability of the face and the deformations occurring around the tunnel were examined. The section of Km: 216 + 580 was considered for the analysis (see Figure 4). Here, the overburden thickness is 72 m, and the tunnel was completely excavated in the graphitic schists. The graphitic schists are characterized as dark gray, weak to very weak, and medium weathered. In this section, the geotechnical parameters are given in Table 1.

**Table 1.** Rock mass parameters of the graphitic schists [36].

| Uniaxial Comp. Strength UCS (MPa) | Geological Strength Index GSI | Material Constant $m_i$ | Elastic Modulus $E_i$ (MPa) | Unit Weight $\gamma_n$ (kN/m³) | Cohesion c (kPa) | Internal Friction Angle $\phi$ (°) | Deformation Modulus $E_m$ (MPa) |
|---|---|---|---|---|---|---|---|
| 27 | 25 | 10 | 4500 | 22.0 | 240 | 37 | 270 |

In the first stage, three-dimension (3D) analyses were performed employing the existing rock mass parameters, and then the analyses were repeated using different rock mass parameter values. Three-dimensional numerical analyses are useful to understand all stress and deformation distributions around structures such as tunnels, buildings, and bridges [19,49–52]. The created model and boundary conditions are shown in Figure 8a,b. In the model, in-situ stresses are created by using gravity, and the upper ground level is released in the z-direction. The model was accepted as symmetric at the point 0, 0, 0 and the Mohr–Coulomb failure criterion was chosen. The tunnel was excavated in one level and immediately defined as the segment covering the "shell" element in the model (Figure 8c). The FLAC3D program uses bulk modulus and shear modulus. These values are related to the deformation modulus and the Poison ratio [48]. The bulk modulus and shear modulus are calculated as 225 MPa and 103 MPa, respectively.

In the analyses, the Y- and Z-axis deformations are investigated; the Z-direction refers to the vertical deformations, in addition, the Y-direction also reflects the tunnel face deformations. When the deformations in the tunnel are examined, the deformations in the Z-direction (vertical direction) remained at the level of 2 cm. (Figure 9a). In addition, displacements in the Y direction (in the tunnel face) are up to 6 cm (Figure 9b). This situation reveals the result that tunnel face stability should be provided because a displacement of up to 6 cm in the tunnel face is critical. However, since single-shield TBM was selected, this situation in tunnel face stability was ignored. Therefore, a continuous flow of material occurred in the tunnel face during the excavation. In this case, it resulted in a relaxation zone on the top of the tunnel. This situation caused extra loads on the segments and caused the segment failures. To prevent these problems, revisions were made to the TBM and the thrust of the TBM was increased, and the EPBM type was adopted for face stability.

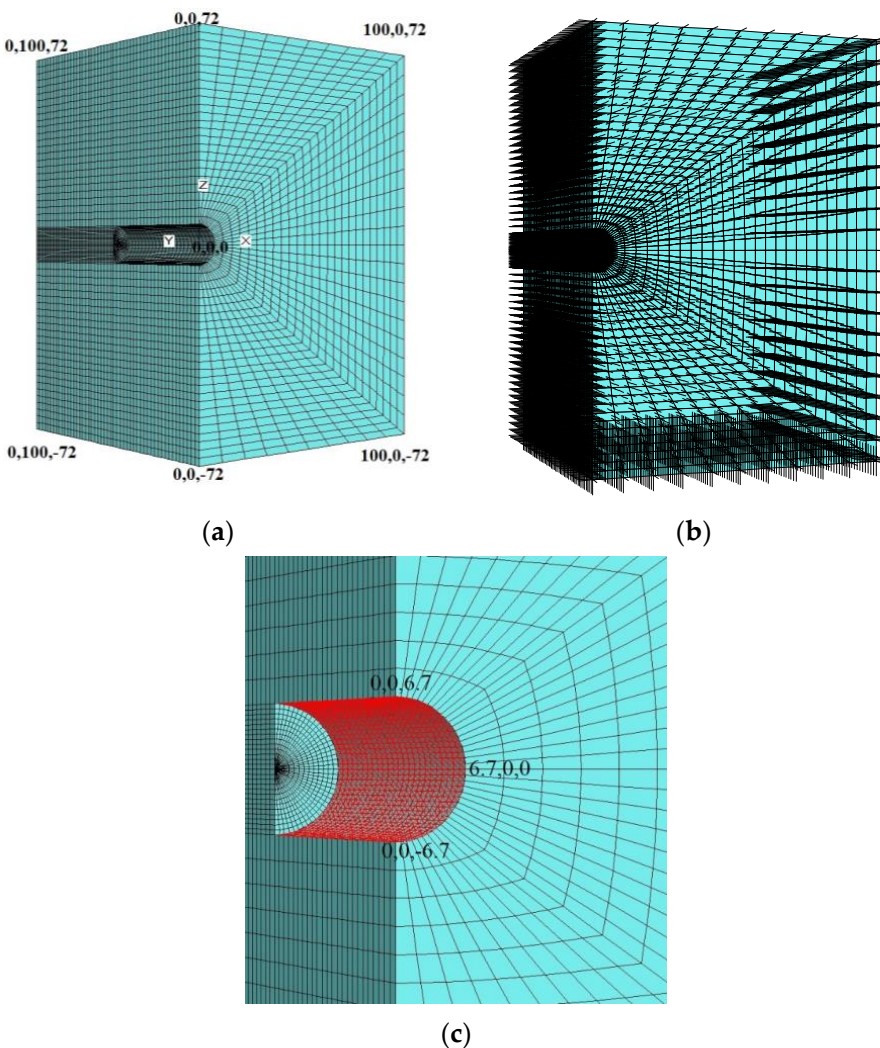

**Figure 8.** (**a**) FLAC3D model; (**b**) FLAC3D model boundary conditions; (**c**) FLAC3D model tunnel section and shell element.

Back analysis was performed using different parameters to model the failure affecting the tunnel. Here, the analysis was repeated considering very low cohesion because the material comes from the tunnel face continuously. For this case, analyses were performed for different deformation modulus, cohesion, and internal friction angles to determine the suitable ground parameters in the tunnel section (Table 2) with the same segment support properties. These parameters are selected considering previous studies, laboratory test results, and site investigations.

The deformation values obtained from the analyzes for three different parameter sets are given in Table 3 and Figure 10a–f. As can be seen from the results, the displacements increase as the deformation modulus and cohesion value decrease. The displacements in the Y-direction of up to 3 m in the tunnel face were determined in Analysis 3.

The flow of material that occurred in the tunnel face also affected the excavated parts of the tunnel. Consequently, segments have become unable to carry this overburden. After these stages, although the type of TBM was converted to EPB, the power of the machine was increased and the segment design was changed. The TBM was stuck and the tunnel excavation was stopped due to the cohesionless behavior of the material on the tunnel without the effect of arching.

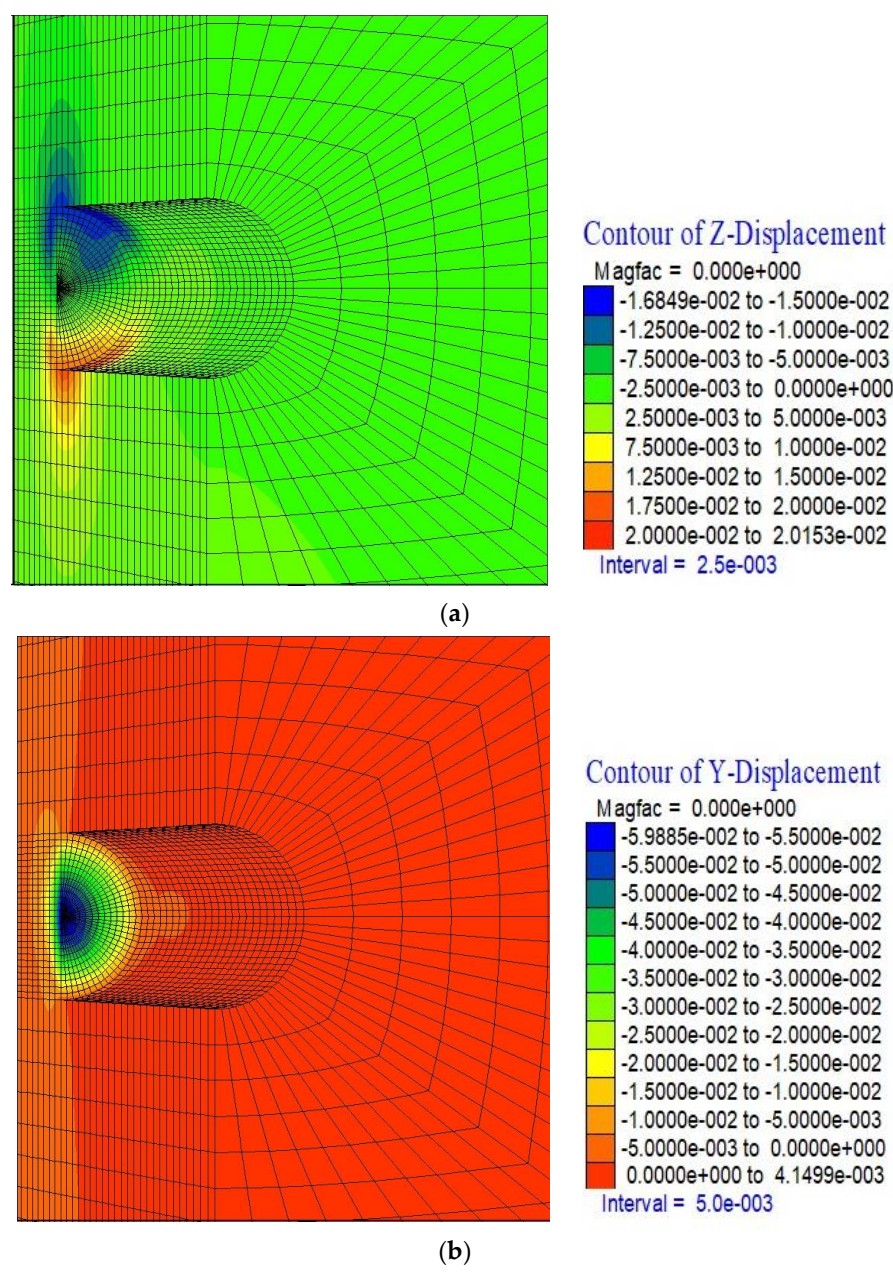

**Figure 9.** Deformations in Z-direction (**a**) and Y-direction (**b**). (units in m).

**Table 2.** Rock mass parameters employed during the back analyses.

|  | Deformation Modulus (MPa) | Cohesion (kPa) | Internal Friction Angle (φ) | Bulk Modulus (MPa) | Shear Modulus (MPa) |
|---|---|---|---|---|---|
| Analysis-1 | 270 | 100 | 15 | 225 | 103 |
| Analysis-2 | 150 | 50 | 15 | 125 | 58 |
| Analysis-3 | 100 | 20 | 15 | 83 | 38 |

**Table 3.** Rock mass parameters employed during the back analyses.

|  | Displacement in Y-Direction | Displacement in Z-Direction |
|---|---|---|
| Analysis-1 | 10 cm | 2.42 cm |
| Analysis-2 | 34 cm | 9.8 cm |
| Analysis-3 | 300 cm | 223 cm |

### 4.5. Analysis of TBM Section under Squeezing Conditions

As stated by Ramoni and Anagnostou [53], TBM performance is the result of a complex interaction between the ground, the tunneling equipment (TBM and backup) and the support. For this reason, in this section, the results of the analysis are evaluated according to the determined back analysis method. For the TMB section, the thickness of the segments is 45 cm, and the concrete class is taken as C40/50. In the analysis, it is assumed that the first 40 m section of the tunnel is excavated. Afterward, the deformations are reset, and tunnel excavation is started in 2 m stages. Tunnel stability is examined in nine stages of excavation between the 40th and 54th m in the tunnel. The segment lining is defined as the shell element in the model. The segment lining is installed in the model after each excavation step. The segment properties are given in Table 4 and the segments placed in the model are presented in Figure 11 and the modeling stages are presented in Table 5. The deformation results obtained from the analyses are given in Table 6.

As can be seen from the results of the analysis, a significant increase is observed in the deformations that occur in the tunnel due to the tunnel excavation (Table 6). Between the 40th and 42nd m of the first excavation, the vertical deformations on the tunnel ceiling are 76 cm, the horizontal displacements are 8.56 cm, and the deformations on the tunnel face are 126 cm. In a sense, serious deformations occur in the tunnel when excavation is started without providing tunnel face stability on such ground. At the end of the 12 m excavation in the tunnel, the deformations that occur are 306 cm on the tunnel ceiling, 80.1 cm horizontal displacements, and 385 m deformations on the tunnel mirror. As can be seen, these values show that the tunnel is completely squeezing and that the tunnel face and ceiling stability cannot be achieved. Considering the surface settlements encountered in the tunnel and the failures of the support systems, the analyses reflect the field conditions. This confirms both the sinking of the TBM and the compression in the tunnel. Figure 12 shows the deformations in the vertical, horizontal, and tunnel faces that occurred when the tunnel excavation started; in Figure 13, the deformations occurred when the tunnel excavation progressed 12 m. After the first excavation and 12 m excavation, vertical deformations increased four times, horizontal deformations increased nine times and tunnel face deformations increased three times. It is seen that tunnel excavation is not possible without providing a tunnel ceiling and face stability.

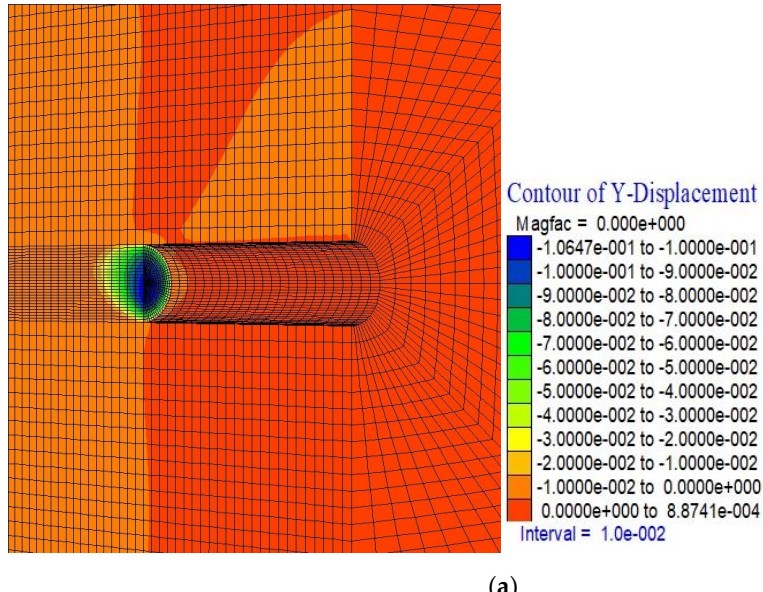

(**a**)

**Figure 10.** *Cont.*

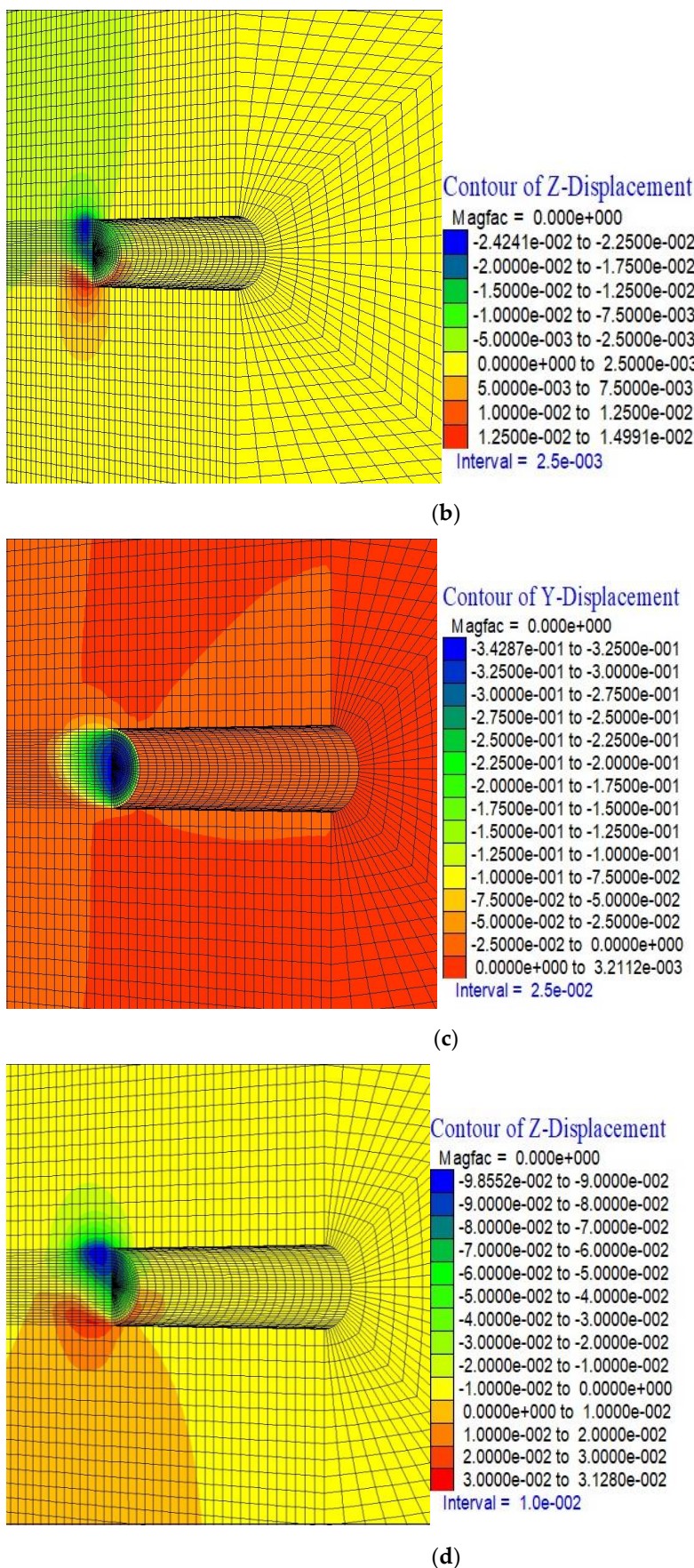

**Figure 10.** *Cont.*

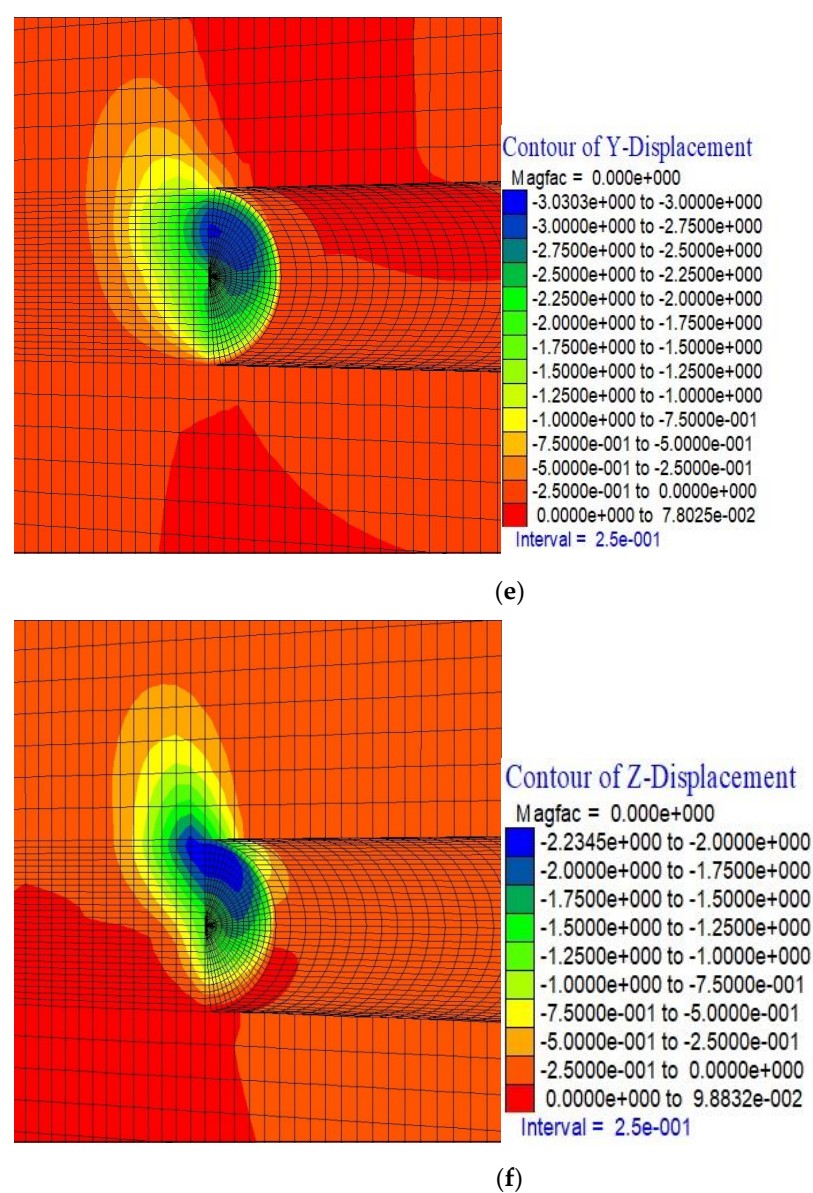

(**e**)

(**f**)

**Figure 10.** Displacements in Y-direction for the Analysis-1 (**a**); displacements in Z-direction for the Analysis-1 (**b**); displacements in Y-direction for the Analysis-2 (**c**); displacements in Z-direction for the Analysis-2 (**d**); displacements in Y-direction for the Analysis-3 (**e**); displacements in Z-direction for the Analysis-3 (**f**) (units in m).

**Table 4.** Parameters used for segment lining.

| Element | $E_i$ (GPa) | $\sqrt{}$ | $\Upsilon$ (kg/m3) | ds (cm) | fck (MPa) |
|---------|-------------|-----------|---------------------|---------|-----------|
| Segment Lining | 34 | 0.25 | 2500 | 45 | 40 |

As can be seen from the results of the analysis, a significant increase is observed in the deformations that occur in the tunnel due to the tunnel excavation (Table 6). Between the 40th and 42nd m of the first excavation, the vertical deformations on the tunnel ceiling are 76 cm, the horizontal displacements are 8.56 cm, and the deformations on the tunnel face are 126 cm. In a sense, serious deformations occur in the tunnel when excavation is started without providing tunnel face stability on such ground. At the end of the 12 m excavation in the tunnel, the deformations that occur are 306 cm on the tunnel ceiling, 80.1 cm horizontal displacements, and 385 m deformations on the tunnel mirror. As can be seen, these values show that the tunnel is completely squeezing and that the tunnel face and

ceiling stability cannot be achieved. Considering the surface settlements encountered in the tunnel and the failures of the support systems, the analyses reflect the field conditions. This confirms both the sinking of the TBM and the compression in the tunnel. Figure 12 shows the deformations in the vertical, horizontal, and tunnel faces that occurred when the tunnel excavation started; in Figure 13, the deformations occurred when the tunnel excavation progressed 12 m. After the first excavation and 12 m excavation, vertical deformations increased four times, horizontal deformations increased nine times and tunnel face deformations increased three times. It is seen that tunnel excavation is not possible without providing a tunnel ceiling and face stability.

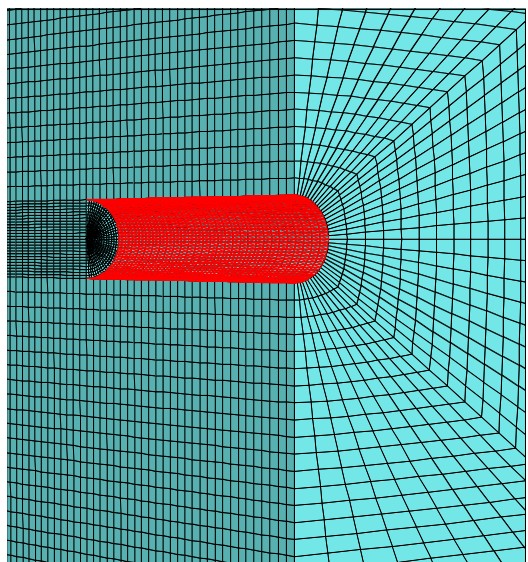

**Figure 11.** Tunnel segment lining defined as shell elements.

**Table 5.** Excavation stages.

| Stages | Phase |
|---|---|
| 1 | Initial stresses |
| 2 | Reset of the displacements |
| 3 | Excavation and installation of the segments between 40 to 42 m |
| 4 | Excavation and installation of the segments between 42 to 44 m |
| 5 | Excavation and installation of the segments between 44 to 46 m |
| 6 | Excavation and installation of the segments between 46 to 48 m |
| 7 | Excavation and installation of the segments between 48 to 50 m |
| 8 | Excavation and installation of the segments between 50 to 52 m |
| 9 | Excavation and installation of the segments between 52 to 54 m |

**Table 6.** Deformation results.

| Stage | Vertical Displacement (cm) | Horizontal Displacement (cm) | Face Displacement(cm) |
|---|---|---|---|
| 3 (Between 40 to 42 m) | 76 | 8.56 | 126 |
| 4 (Between 42 to 44 m) | 149 | 26.8 | 221 |
| 5 (Between 44 to 46 m) | 199 | 45.3 | 288 |
| 6 (Between 46 to 48 m) | 251 | 61.2 | 335 |
| 7 (Between 48 to 50 m) | 285 | 71.7 | 365 |
| 8 (Between 50 to 52 m) | 304 | 77.5 | 381 |
| 9 (Between 52 to 54 m) | 306 | 80.1 | 385 |

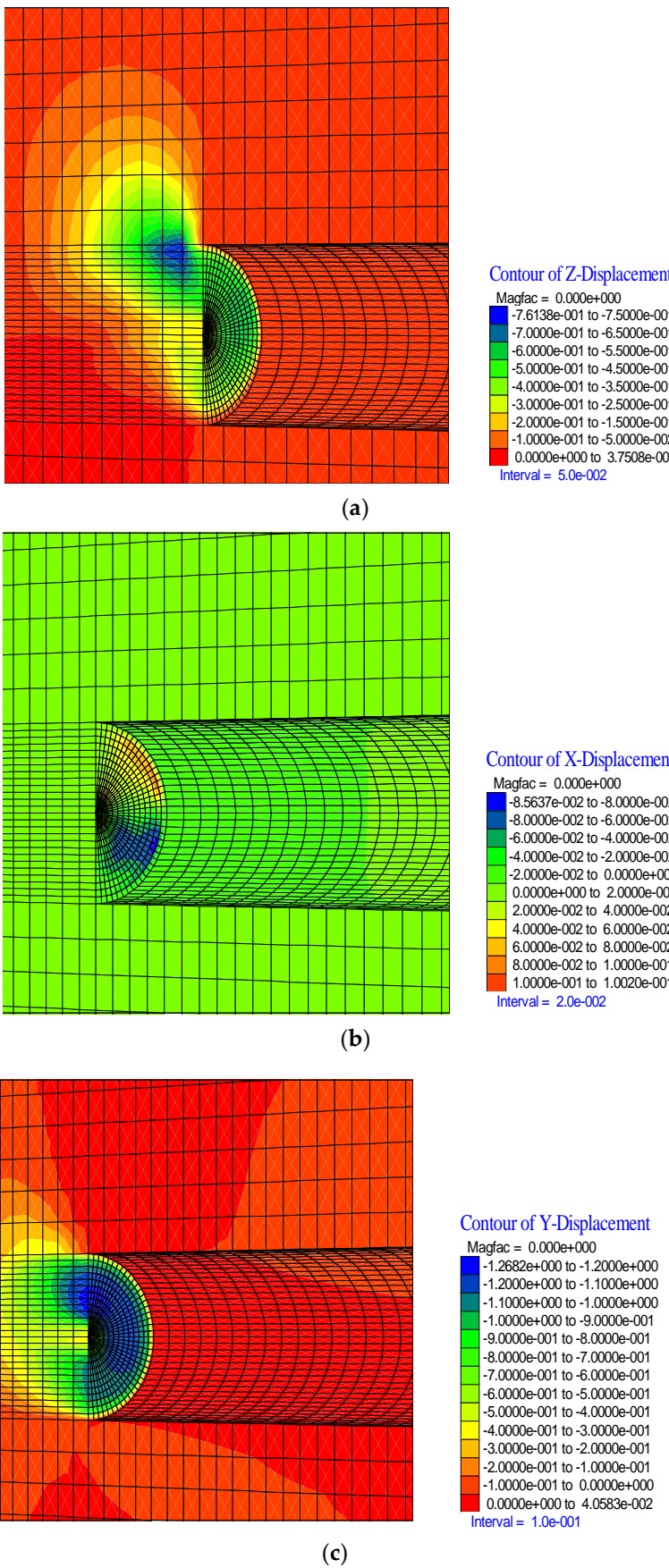

**Figure 12.** Deformation results in third stage (**a**) Vertical displacement, (**b**) Horizontal displacement, (**c**) Face displacement. (units in m).

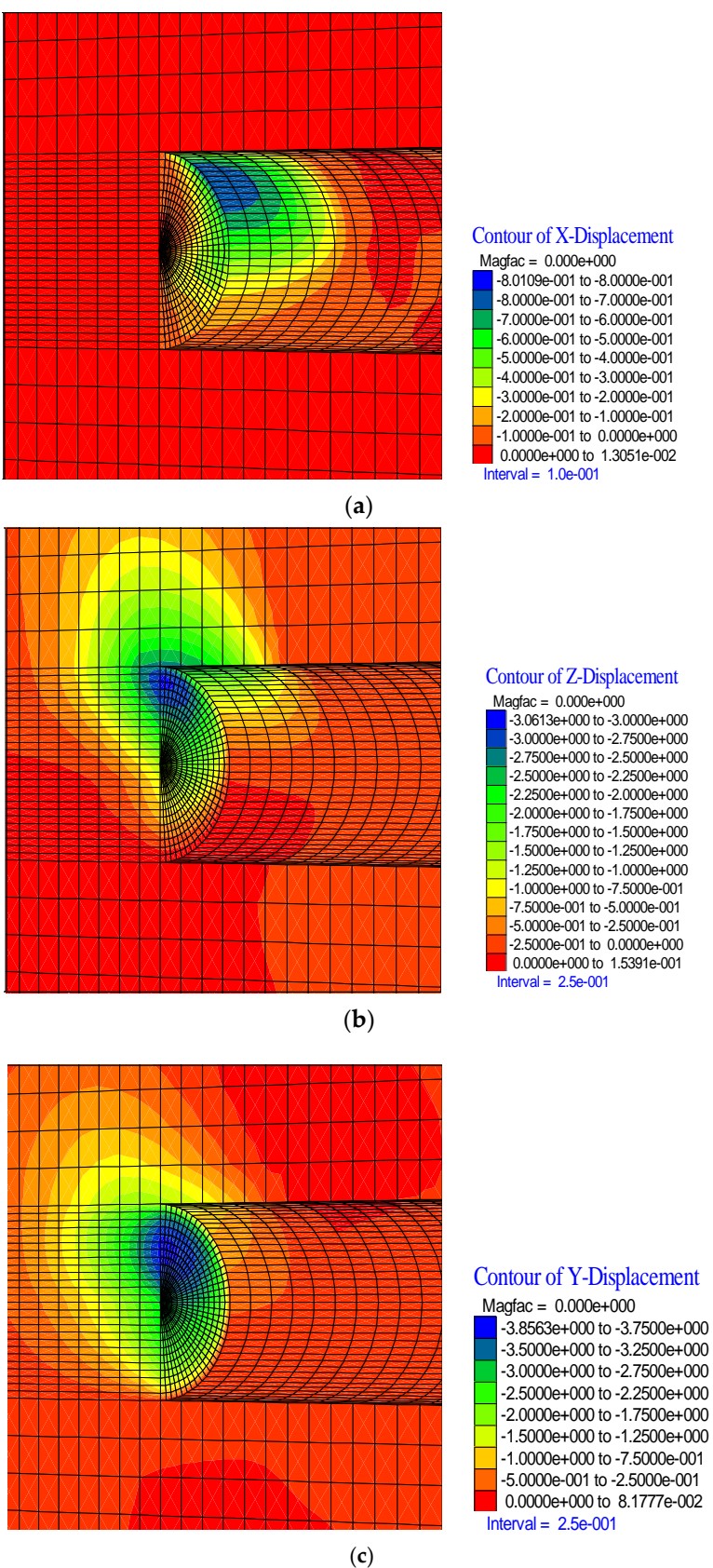

**Figure 13.** Deformation results in eighth stage (**a**) Vertical displacement, (**b**) Horizontal displacement, (**c**) Face displacement (units in m).

### 4.6. Analysis for TBM Section under Squeezing Conditions

The most important question to be answered to construct the T26 tunnel is "what is the relationship between the failure surface of the large landslides along the tunnel route and the tunnel location?". To answer this question, an extensive field study was performed to describe the failure surface of the landslides on the tunnel route. Five separate landslide masses were detected during the field investigations. These landslides were evaluated by using a 1/25,000 scale digital elevation model and the subsequent field studies were carried out (Figure 14). It was thought that the main failures among these were composed of the landslides named Ls-2 having an area of approximately 703,000 m$^2$ and Ls-4 having an area of approximately 466,000 m$^2$. On the other hand, the Ls-1 having an area of approximately 96,000 m$^2$ and Ls-3 having an area of approximately 132,000 m$^2$ are observed as the secondary landslides developed inside the mass of Ls-2. The last position of the TBM on the tunnel route is located underneath the landslide Ls-2. Larsen et al. [54] suggest the following relation between the landslide surface area and volume:

$$V = \alpha A^{\gamma} \qquad (1)$$

where V is the volume of the landslide (m$^3$) and A is the surface area of the landslide (m$^2$). Larsen et al. [54] state that landslide volumes can be calculated at R$^2$ = 0.95 explained variance level by using log$\alpha$ = −0.836 ± 0.015 and $\gamma$ = 1.332 ± 0.005 coefficients. Following the calculation of the landslide volumes, the depths of the failure surfaces were calculated by considering the landslide dimensions suggested by IAEG [55] and WP/WLI [56]. The approximate volume relating to the Ls-1 is estimated to be 629,000 m$^3$ while the estimated depth of the failure surface was calculated as approximately 40 m. However, the approximate volume of the Ls-2 mass that will be passed through along the route was estimated as 8,950,000 m$^3$ and the approximate depth of the failure surface was calculated as 77 m (Figure 14a). The volumes calculated for the Ls-4 and Ls-5 (Figure 14b) and the estimated depths of the failure surfaces are presented in Table 7. These values were calculated empirically assuming that they comply with the area of the landslides and the circular failure model. For this reason, these results were compared first with the surface and inclinometer measurements and then interpreted.

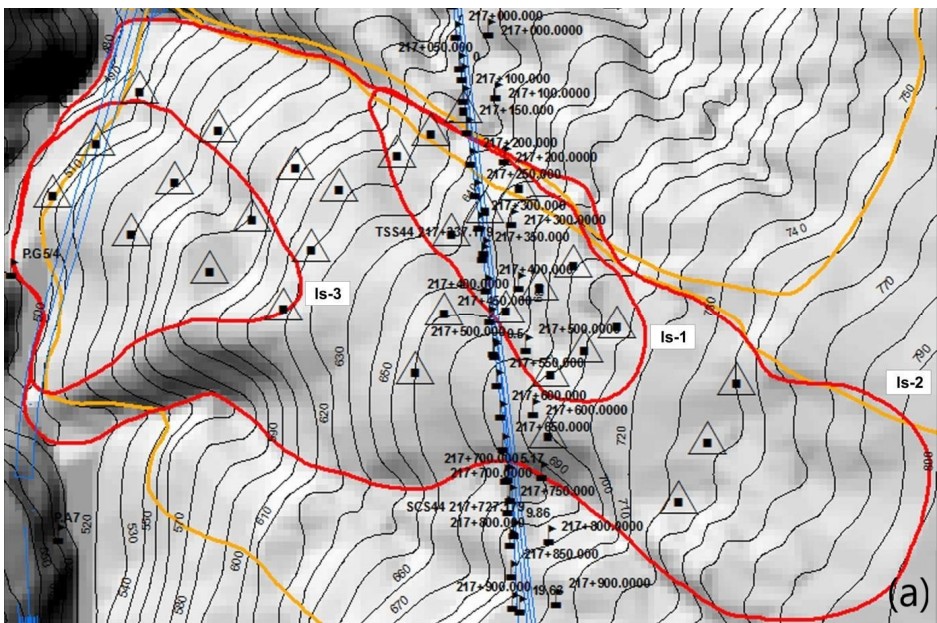

**Figure 14.** *Cont.*

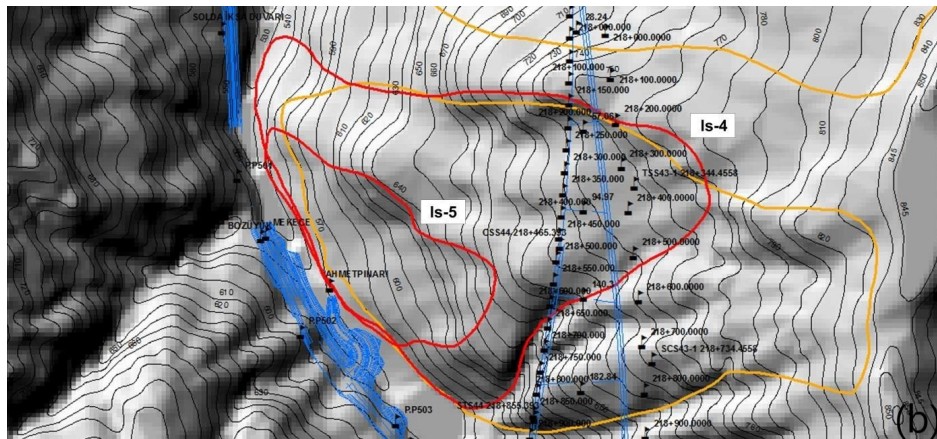

**Figure 14.** Areal distributions of the landslides Ls-1, Ls-2, and Ls-3 interpreted on the 1/25,000 scale digital elevation model; the spatial distribution and the positions of the monitoring stations; 30 monitoring stations are indicated by points in triangle symbols (**a**); areal distributions of the landslides Ls-4 and Ls-5 interpreted on the 1/25,000 scale digital elevation model (**b**).

**Table 7.** Area (m$^2$) and volume (m$^3$) values calculated relating to the landslides distinguished from the 1/25000 digital elevation model along the tunnel route, and the depths of the failure surfaces estimated empirically.

| Landslide | Area (m$^2$) | Volume (m$^3$) | Estimated Deepest Failure Surface Depth (m) |
|---|---|---|---|
| Ls-1 | 95658 | 629,033.28 | 39.46 |
| Ls-2 | 702,216 | 8,950,593.95 | 76.48 |
| Ls-3 | 131,957 | 965,536.86 | 43.90 |
| Ls-4 | 466,271 | 5,187,752.06 | 66.76 |
| Ls-5 | 120,715 | 857,548.76 | 42.62 |

A total of 30 surface measurement stations were established to monitor the surface deformations that occurred due to the landslides observed through the T-26 tunnel route (Figure 14a). Measurements were taken by using differential GPS from the stations in the period of 24 February 2013 and 29 May 2013. The first measurements were more frequent. The precisions of the measurements in the x, y, and z directions were assigned to be 1 mm. Total deformations determined in the x, y, and z directions depending on the measurements taken from the stations are given in Figure 15. Accordingly, the highest resultant deformation in the period of measurement was calculated as 8.3 cm at station No. 17. The areal distribution of the resultant movements detected at the stations is presented in Figure 16a. As can be seen in Figure 16a, the maximum resultant movement vectors were detected at stations No. 7 and 17, which were close to these locations.

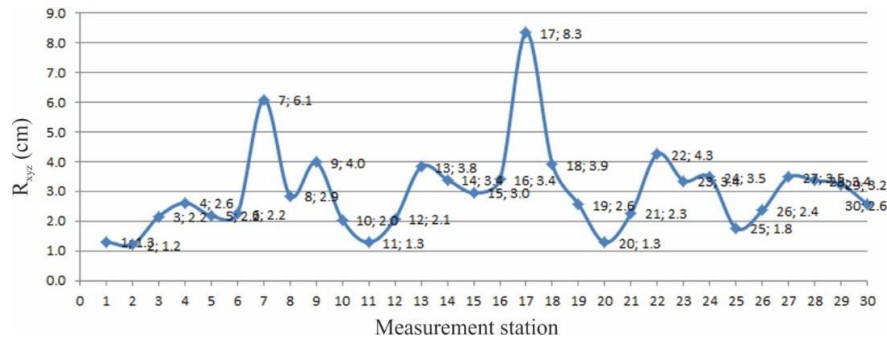

**Figure 15.** Resultant movement vector magnitudes (cm) calculated according to the measurements taken from the monitoring stations in the period of 24 February 2013 and 29 May 2013.

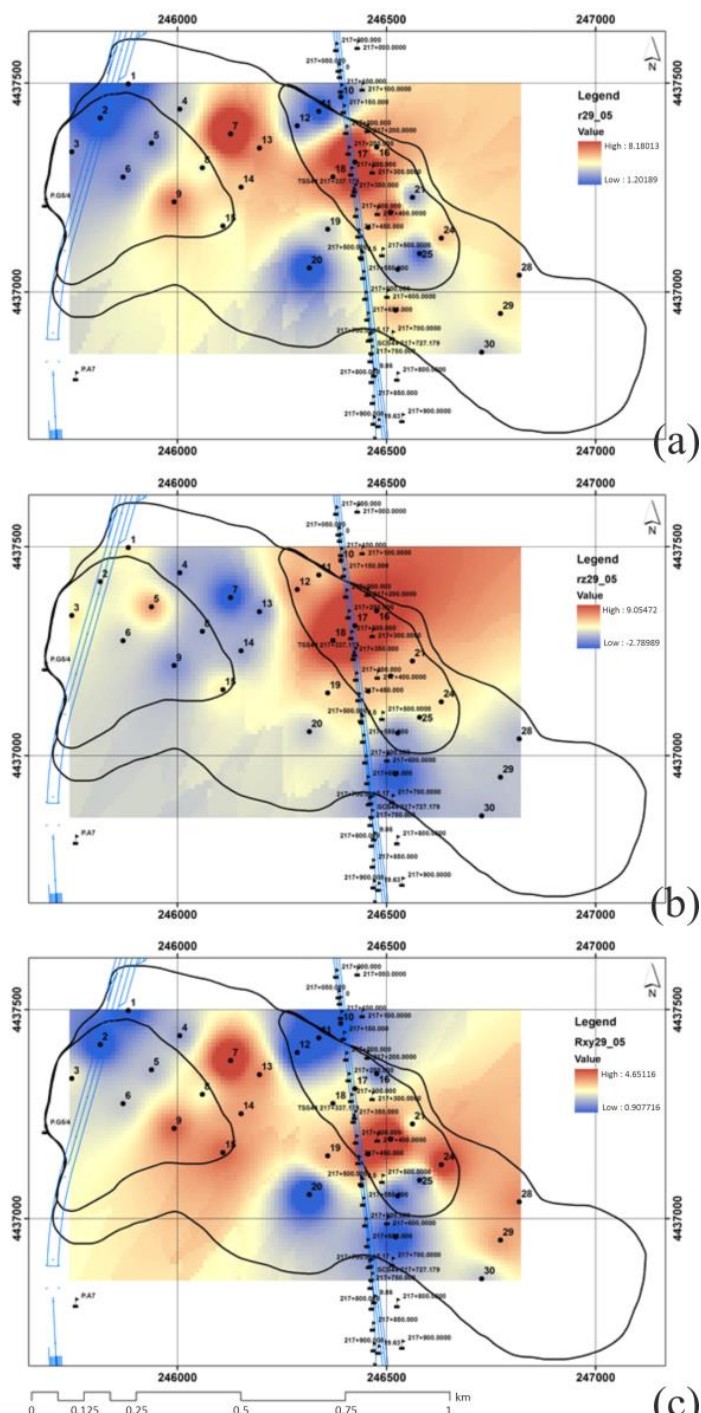

**Figure 16.** Areal distribution of the resultant movement vector magnitudes (cm); the measurements were taken from the monitoring stations in the period of 24 February 2013 and 29 May 2013 (**a**); areal distribution of the vertical movement vector magnitudes (cm) calculated according to the measurements taken from the monitoring stations in the period of 24 February 2013 and 29 May 2013 (**b**); areal distribution of the horizontal movement vector magnitudes (cm) calculated according to the measurements taken from the monitoring stations in the period of 24 February 2013 and 29 May 2013 (**c**).

To better understand the mechanisms of the landslides and their relationship with TBM, the areal distributions of the movements both in the horizontal and vertical directions were determined separately. The distribution of the movement in the vertical direction is shown in Figure 16b. Accordingly, downward vertical deformations reaching up to 9 cm were detected along the tunnel route and particularly in the section where the TBM operates. However, upward vertical movements were detected in measuring locations No. 4, 7, 8, and 9. The main reason for the occurrence of the upward movements can be explained by the landslide Ls-2, particularly by tilting and heaving in the toe section of the failure. The areal extension of the vectors in the vertical direction (Figure 16b), stigmatic surfaces, and pit formations in the landslide are closely related to topographic hills and depressions which can be evaluated as typical hummocky style topography observed in landslides. As a consequence, the results obtained are reliable and reflect the typical landslide topography. The areal distribution of the vertical movement vector magnitudes (cm) calculated according to the measurements taken from the monitoring stations in the period between 24 February 2013 and 29 May 2013 are given in Figure 16c. Accordingly, a horizontal movement that can be described as almost linear in the northern part of the field and on stations No. 7 and 24 is observed. Within the period in which the movements were detected during the measurements, TBM had been operating under this section. Therefore, there should be a relationship between the operation of TBM and the slope movements. This situation was considered as the re-activation of the slope by additional effects created in the landslide region. The directions of the vectors determined for the stations having an amount of total horizontal movement greater than 3 cm are given in Figure 17a. The measurements for the resultant vectors given in Figure 17a were taken between the same dates 24 February 2013 and 29 May 2013. Accordingly, the direction of the movements is downward. The maximum horizontal movement was calculated as 4.7 cm. This value was obtained from the measurements performed on station No. 22. Station No. 22 is located on the surface of the section where the TBM was operated for the last time. Even though the maximum horizontal movement was determined in this section, no deformation was encountered in the tunnel. This is one of the important findings indicating that the tunnel is at a deeper level than the failure surface of the landslide. The locations of the movement vectors obtained from the monitoring stations on the cross-section are shown in Figure 17b. It is expected that the angle of the surface deformations with the horizontal should be compatible; accordingly, the obtained angles are in accordance with the surface of the failure of the surface deformation vectors. However, as can be seen from Figure 17b, the surface deformation measurements showed parallelism mainly with the predominantly shallow surface of failure ls-1. This is a physically normal situation. The total amount of movement obtained from the surface vectors indicates the maximum movement of both landslides.

To investigate the landslide-tunnel relationship, in addition to empirical approaches and in-situ measurements, two- and three-dimensional (2D and 3D) numerical analyses were also performed. The section at Km: 214 + 400 was chosen for analysis (see Figure 4); because, as seen in Figure 16a, there is a landslide on the surface and its depth and geometry were determined. The cross-section and the parameters used in the analyses are given in Figure 18. In the first stage, to assess the stability of the landslide detected on top of the T26 tunnel, 2D numerical analyses were performed via Phase 2D v8.0 software with a shear strength reduction (SSR) approach that systematically reduced the shear strength envelope of material by a factor of safety [57]. Phase2D v8.0 is used to determine the safety factor of a simple homogeneous slope using the shear strength reduction (SSR) method (Figure 19a). As can be seen from the slope stability analysis (Figure 19b), the critical SRF was obtained as 1.27 and the slope movement does not affect the tunnel.

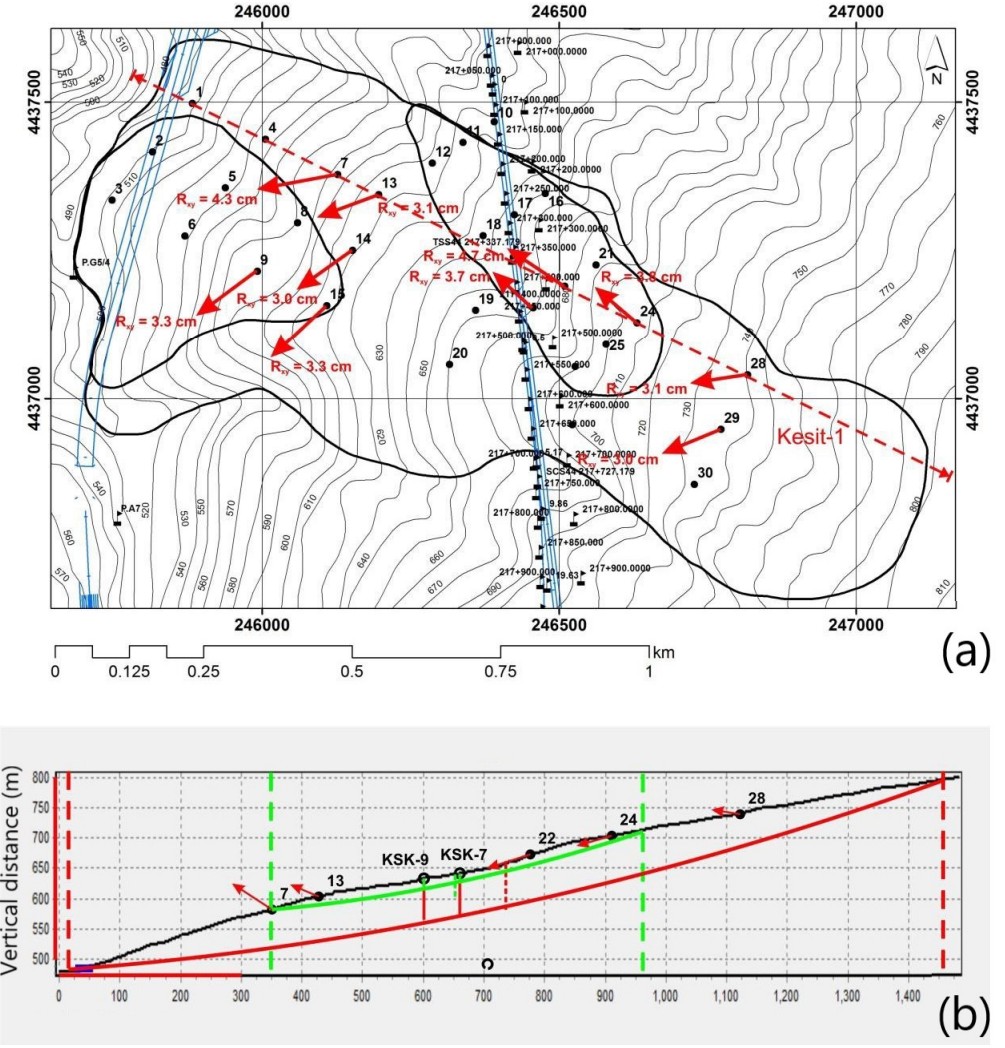

**Figure 17.** Directions of the vectors determined for the stations having the amount of total horizontal movement greater than 3 cm; the measurements were taken between the dates 24 February 2013 and 29 May 2013 (**a**); locations of the movement vectors obtained from the monitoring stations on the cross-section; the colors green and red represent Ls-1 and Ls-2, respectively; the tunnel is also represented (**b**).

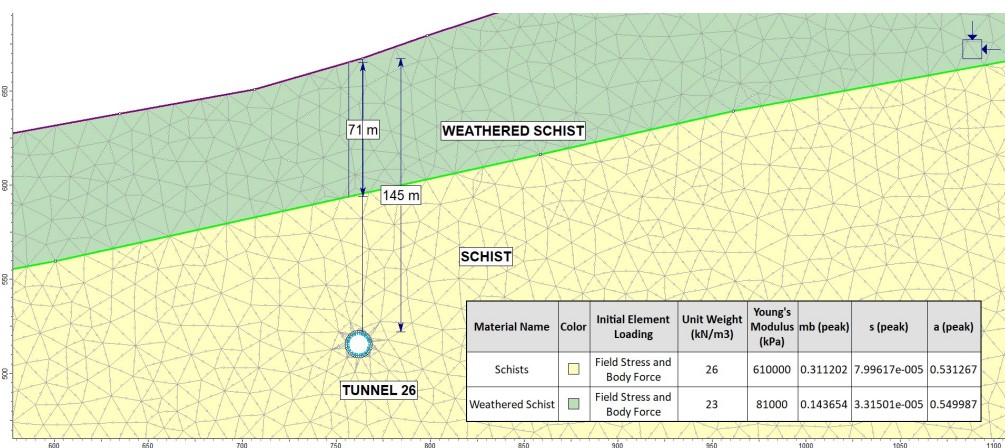

**Figure 18.** The geological cross-section used in 2D numerical analysis and the material properties.

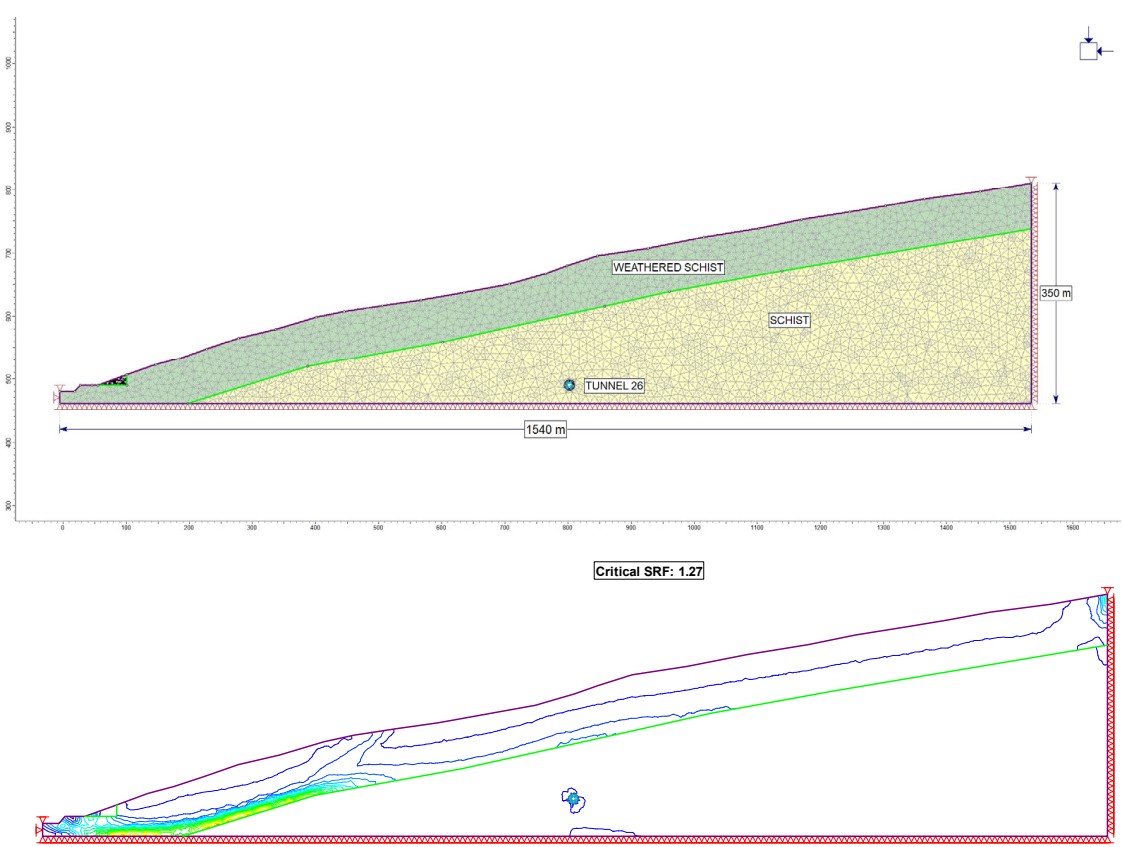

**Figure 19.** Phase2D model for the slope stability analysis (SRF analysis) and the result of slope stability analysis.

In addition, for the tunnel excavation and support system, 2D numerical analyses were performed with the Phase2D v8.0 program and 3D analyses with the FLAC3D program. The model created for 2D analysis with the Phase2D program is given in Figure 20. Analyses were performed in three stages. In the first stage, the field stresses were described, the tunnel excavation was in the second stage, and immediately the segments were placed, and in the third stage, the seismic parameters were described in the model. The total deformation in the tunnel was calculated as 1.65 cm in the analysis. Likewise, the same deformations under seismic conditions were obtained. The main reason for this is that the overburden thickness is more than 100 m (Figure 21a,b).

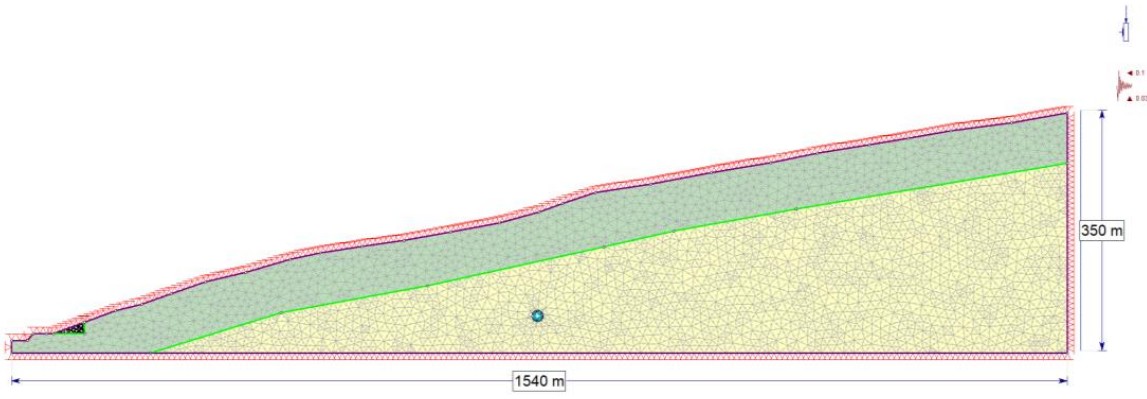

**Figure 20.** Boundary conditions of 2D model for determining the tunnel excavation and support systems.

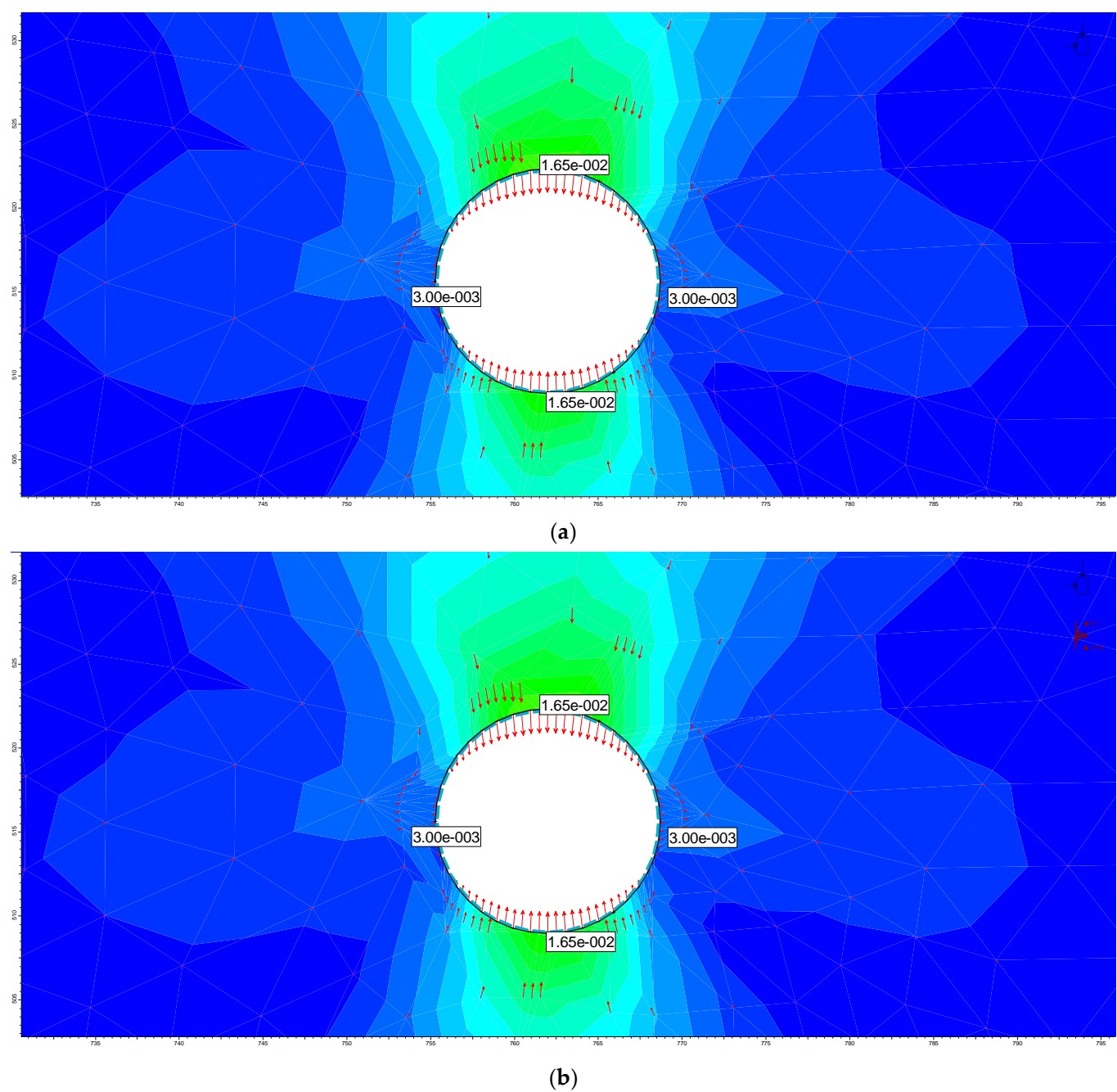

**Figure 21.** Total displacements obtained from the 2D numerical analysis (**a**) and total displacements obtained from the 2D numerical analysis (seismic conditions) (**b**). (units in m).

Three-dimensional numerical analyzes were performed with the FLAC3D program, and the same result was found with 2D numerical analyzes, in other words, 1.7 cm total deformation was obtained (Figure 22a). In addition, deformations occurring in the tunnel face were also investigated, and 7 cm deformation was calculated (Figure 22b). This situation reveals that the face stability of the tunnel must be maintained, continuously.

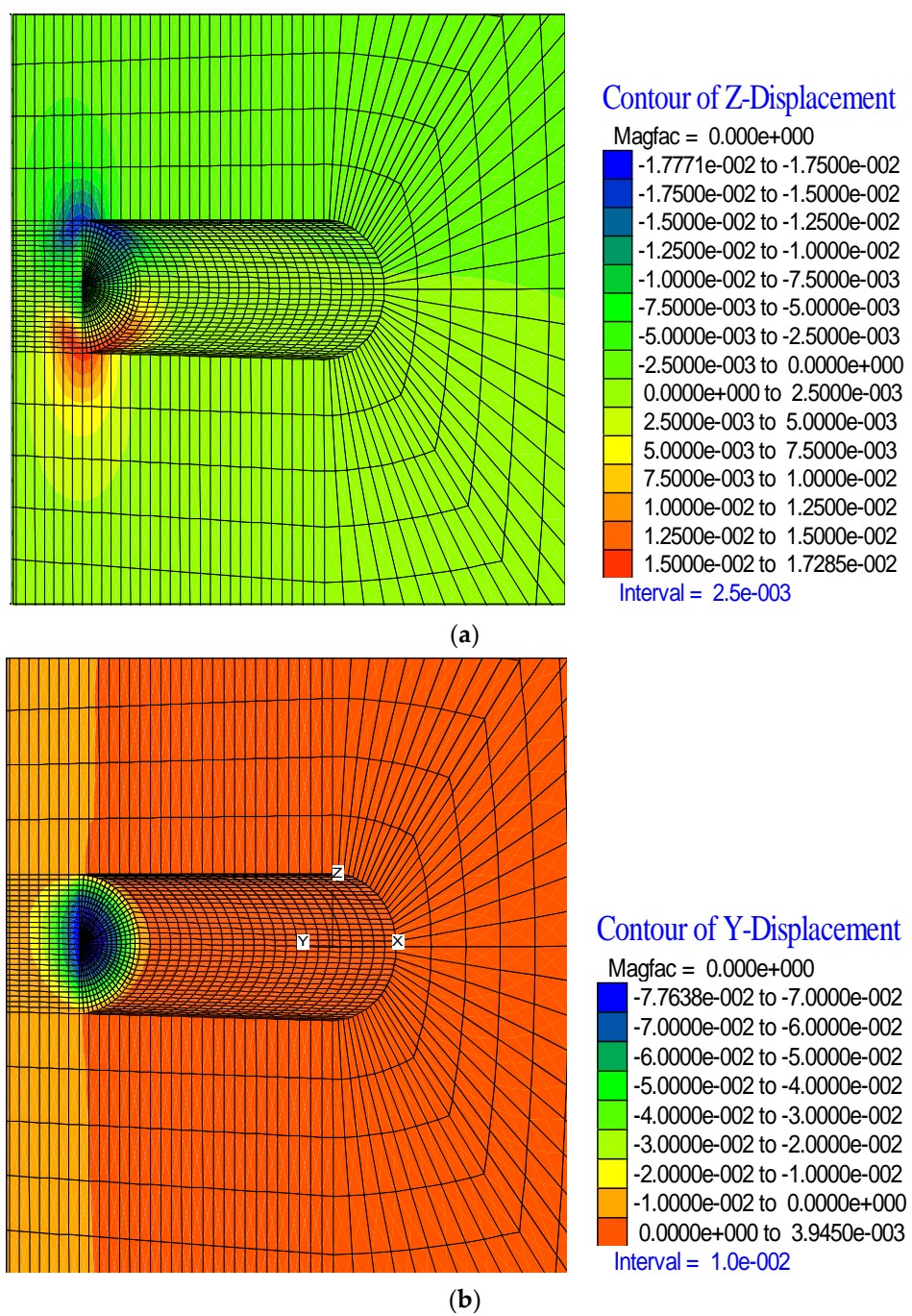

**Figure 22.** Vertical deformations (**a**) and horizontal deformations (**b**) obtained from the 3D numerical analyses (units in m).

## 5. Discussions and Conclusions

To solve the problems and indicate the conditions of the landslide on the tunnel route, depending on the analyses performed, it was identified that the depth of the failure surface is 80 m below the tunnel. However, the TBM was completely stuck at Km: 217 + 526, and the excavations were terminated. In addition, both 2D and 3D numerical analysis results also did not show a clear relationship between excessive deformations in the tunnel and the landslide. Therefore, a direct relationship between landslide and TBM jamming could not be detected.

As a consequence, in July 2011, the excavations of the T26 tunnel were initiated by TBM. At first, the 1020 m part of the tunnel was completed. However, deformations that emerged inside the tunnel resulted in subsidence on the surface and failures on the linings. The excavations proceeded more than 246 m; however, despite the additional reinforcements, the TBM was stuck due to the extreme deformations that occurred at Km: 217 + 526. Consequently, the excavations were interrupted for the tunnels excavated through grounds representing weak rock characteristics and squeezing ground features. For this reason, the TBM type must be selected as Earth Pressure Balance (EPB). Those types of ground conditions were encountered in the T26 tunnel. As a result of the face stability problems during the tunnel excavation, the TBM could not advance; however, over-excavation was carried out by continuously taking the material from the tunnel face, and the subsidence at the surface formed. TBM was revised as a result of these problems. Nevertheless, the problems experienced caused the ground around the tunnel to plastic zone completely, and the plastic zone reached the surface. Although both the TBM and segments were revised after this stage, these efforts were not sufficient and TBM lost its function completely. At this stage, considering the resultant conditions, it was decided to continue excavation by the NATM method. Subsequently, tunnel excavation was interrupted once again. Nevertheless, re-design studies of the T26 tunnel were initiated in 2016 and then, Fugro Sial [36] conducted relevant studies for the design phase according to NATM. Following that, the tunnel excavation was finally re-started in 2017 and it has been successfully proceeding with NATM.

Numerous problems were repetitively encountered since the beginning of the TBM excavations. High-stress impacts were observed on the tunnel sections due to the deformations that occurred during these stability problems so continuous reinforcement applications were carried out. Considering the resultant situation, the re-design stage was initiated according to the NATM method. Following 2016, the TBM was reached by the means of excavations proceeded concerning the NATM method. Segment linings were completely collapsed; segment blocks had overlapped each other. As a result, segments became unable to bear loads. Briefly, this indicated that the thickness of the lining and the amount of reinforcements were insufficient. Considering these failures, the main reasons for the problem can be given as follows: (i) The overburden load which has approximately 130 m thickness became active and arched inside the tunnel, and (ii) both the existence of active landslides and the occurrence of subsidence on the surface during the TBM excavations.

For the excavations through those types of ground conditions, it is critical to decide between TBM and NATM methods. The advantages of preferring the NATM method can be concluded as follows: (i) The advantages of interfering in tunnel support systems instantaneously, adapting and changing support types depending on the ground conditions, (ii) revising support systems and implementing specific support configurations in compliance with the encountered ground types. On the other hand, the TBM must be selected as an EPB type, if the ground represents weak rock characteristics and swelling-squeezing features. However, in the case of encountering unexpected ground conditions, modification and adaptation of the TBM type can be ineffective. Unfortunately, these situations may result in evacuating the tunnel without dislodging the TBM outside the tunnel.

Consequently, the experiences gained from the study are briefly summarized below. These are extremely important for practical tunnel engineering.

(a) Unexpected conditions are likely to occur in extremely complex geological environments. This uncertainty must be taken into account, especially if excavations are applied with TBM in long tunnels to be constructed in extremely complex geological environments.

(b) Especially in high-speed railway projects, construction works must be carried out in adverse geological conditions due to geometric limitations. These issues must be taken into account at the project stage.

(c) If there are landslides or paleo-landslides along the tunnel route, the failure surfaces of these landslides should be determined and their relationship with the tunnel should

be described. In addition, landslides must be taken into account in numerical analysis. The reactivation of paleo-landslides is possible due to tunnel excavations.

(d)     As in the case of T26, high deformations occur, especially due to squeezing in tunnels constructed in extremely weak rock masses. Therefore, the excavation stages should be kept at the shortest possible distance and the ring should be closed and the invert should be completed immediately.

**Author Contributions:** Conceptualization, C.G., H.A.N. and S.G.; methodology, E.B.A. and S.K.; software, E.B.A., H.A.N. and S.K.; validation, S.K. and C.G.; formal analysis, E.B.A. and H.A.N.; investigation, S.K., S.G., C.G.; resources, S.G.; data curation, E.B.A. and H.A.N.; writing—original draft preparation, C.G.; writing—review and editing, C.G. and S.G.; visualization, E.B.A. and H.A.N.; supervision, C.G.; project administration, C.G. and S.G. All authors have read and agreed to the published version of the manuscript.

**Funding:** This research received no external funding.

**Data Availability Statement:** Not applicable.

**Acknowledgments:** The authors would like to thank General Directorate of State Railways of Türkiye (TCDD) and Fugro Sial Geosciences Consulting and Engineering Co for the data support. In addition, the authors would like to thank İçtaş Construction Co for continuous support during the study.

**Conflicts of Interest:** The authors declare no conflict of interest.

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
