# Peer review of "A Geotechnical Perspective on a Complex Geological Environment in a High-Speed Railway Tunnel Excavation (A Case Study from Türkiye)"

_infrastructures, doi:10.3390/infrastructures7110155_

Round 1

Reviewer 1 Report

This manuscript aims to describe the problems encountered during the T26 tunnel and to discuss the sources of the problems. The advantages and disadvantages of TBM and NATM methods for the tunnel having difficult ground conditions were discussed. Based on my review, the manuscript is suitable for publication, where authors applied numerical model and T26 tunnel case. The following comments may help enhancing the quality of this manuscript.

General comments:

[1].       Authors performed good works in this manuscript, but no key contributions can be found. Hence, authors should present the new in this research and the different between this research and current studies. Otherwise, the presented references can guide you to present the main parameters that affects tunnel performance, e.g., (Tunnelling and Underground Space Technology. 123 (2022) 104405. https://doi.org/10.1016/j.tust.2022.104405).

[2].       The information related to the geological conditions of T26 tunnel, the properties of soil, rock characteristics, squeezing ground features, and ground water aquifer are vague and should be clearly added. Otherwise, what are the laboratory and field tests that used in this study?

[3].      3D numerical analyses were carried out with the FLAC3D program, however, the step procedures of the presented numerical model should be clearly discussed. Otherwise, how the authors simulated the excavation process and TBM is this model?

[4].      Authors indicated that despite the additional reinforcements, the TBM was stuck due to the extreme deformations that occurred at Km: 217+526. However, the characteristics of these deformations should be clearly discussed.

[5].      It is necessary to clearly discuss, why the authors specifically preferred NATM instead of TBM in this type of excavation.

[6].      Authors need to present in the discussion section, how this work can be benefit for practical engineering.

[7].      Based on Fig. 6a, the pit dimensions are necessary to be mentioned. Otherwise, the main difference between Figs. 18,19, and 20 should be clarified.

Author Response

The reply file is added.

Reviewer 2 Report

The authors studied the problems encountered during the T26 tunnel and discussed the sources of the problems. The advantages and disadvantages of TBM and NATM methods for the tunnel having difficult ground conditions were presented. The paper is the relevance to the scope of journal. Doubtful or controversial arguments were not detected in the paper. The paper has original content and worthy for publication in the journal. I can recommend it for a possible publication. However, following comments must be considered carefully before this recommendation.

1. In the Introduction part, the authors introduced lots of related researches. However, the problems or limitations of the current researches are not introduced, and the necessity of this research are not explained.

2. The title of the manuscript is like a technical report. The purpose, novelty and investigate method are not reflected.

3. Some figures are not clear enough, and they should be improved.

4. Fig. and Figure are misused in the manuscript.

5. Lines 162: A full stop “.” is missing.

6. Lines 199- 205: “m3” should be changed to be the superscript m3.

7. Line 224: Please elaborate what is NPI 200 type?

8. Line 244: The cause of these problems was the failure of tunnel face stability. Unfortunately, the fundamental literature on face stability is missing. Please add references e.g. 

Effect of inclined layered soils on face stability in shield tunneling based on limit analysis;

Experimental study of the face stability of shield tunnel in sands under seepage condition;

Face stability analysis of circular tunnels driven by a pressurized shield;

Face stability analysis of shallow circular tunnels in cohesive–frictional soils;

Tunnel face stability in cohesion-frictional soils considering the soil arching effect by improved failure models.

9. Figs. 9, 10…: The unit is missing.

10. The conclusions are rather a summary.

Author Response

The reply file is added.
